# A Simple Convergence Proof of Adam and Adagrad

**Alexandre Défossez**                                                   *defossez@meta.com*
*Meta AI*

**Léon Bottou**
*Meta AI*

**Francis Bach**
*INRIA / PSL*

**Nicolas Usunier**
*Meta AI*

**Reviewed on OpenReview:** *https://openreview.net/forum?id=ZPQhzTSWA7*

## Abstract

We provide a simple proof of convergence covering both the Adam and Adagrad adaptive optimization algorithms when applied to smooth (possibly non-convex) objective functions with bounded gradients. We show that in expectation, the squared norm of the objective gradient averaged over the trajectory has an upper-bound which is explicit in the constants of the problem, parameters of the optimizer, the dimension $d$, and the total number of iterations $N$. This bound can be made arbitrarily small, and with the right hyper-parameters, Adam can be shown to converge with the same rate of convergence $O(d \ln(N)/\sqrt{N})$. When used with the default parameters, Adam doesn't converge, however, and just like constant step-size SGD, it moves away from the initialization point faster than Adagrad, which might explain its practical success. Finally, we obtain the tightest dependency on the heavy ball momentum decay rate $\beta_1$ among all previous convergence bounds for non-convex Adam and Adagrad, improving from $O((1 - \beta_1)^{-3})$ to $O((1 - \beta_1)^{-1})$.

## 1 Introduction

First-order methods with adaptive step sizes have proved useful in many fields of machine learning, be it for sparse optimization (Duchi et al., 2013), tensor factorization (Lacroix et al., 2018) or deep learning (Goodfellow et al., 2016). Duchi et al. (2011) introduced Adagrad, which rescales each coordinate by a sum of squared past gradient values. While Adagrad proved effective for sparse optimization (Duchi et al., 2013), experiments showed that it under-performed when applied to deep learning (Wilson et al., 2017). RMSProp (Tieleman & Hinton, 2012) proposed an exponential moving average instead of a cumulative sum to solve this. Kingma & Ba (2015) developed Adam, one of the most popular adaptive methods in deep learning, built upon RMSProp and added corrective terms at the beginning of training, together with heavy-ball style momentum.

In the online convex optimization setting, Duchi et al. (2011) showed that Adagrad achieves optimal regret for online convex optimization. Kingma & Ba (2015) provided a similar proof for Adam when using a decreasing overall step size, although this proof was later shown to be incorrect by Reddi et al. (2018), who introduced AMSGrad as a convergent alternative. Ward et al. (2019) proved that Adagrad also converges to a critical point for non convex objectives with a rate $O(\ln(N)/\sqrt{N})$ when using a scalar adaptive step-size, instead of diagonal. Zou et al. (2019b) extended this proof to the vector case, while Zou et al. (2019a) displayed a bound for Adam, showing convergence when the decay of the exponential moving average scales as $1 - 1/N$ and the learning rate as $1/\sqrt{N}$.

In this paper, we present a simplified and unified proof of convergence to a critical point for Adagrad and Adam for stochastic non-convex smooth optimization. We assume that the objective function is lower bounded, smooth and the stochastic gradients are almost surely bounded. We recover the standard $O(\ln(N)/\sqrt{N})$ convergence rate for Adagrad for all step sizes, and the same rate with Adam with an appropriate choice of the step sizes and decay parameters, in particular, Adam can converge without using the AMSGrad variant. Compared to previous work, our bound significantly improves the dependency on the momentum parameter $\beta_1$. The best known bounds for Adagrad and Adam are respectively in $O((1-\beta_1)^{-3})$ and $O((1-\beta_1)^{-5})$ (see Section 3), while our result is in $O((1-\beta_1)^{-1})$ for both algorithms. This improvement is a step toward understanding the practical efficiency of heavy-ball momentum.

**Outline.** The precise setting and assumptions are stated in the next section, and previous work is then described in Section 3. The main theorems are presented in Section 4, followed by a full proof for the case without momentum in Section 5. The proof of the convergence with momentum is deferred to the supplementary material, Section A. Finally we compare our bounds with experimental results, both on toy and real life problems in Section 6.

## 2 Setup

### 2.1 Notation

Let $d \in \mathbb{N}$ be the dimension of the problem (i.e. the number of parameters of the function to optimize) and take $[d] = \{1, 2, \ldots, d\}$. Given a function $h : \mathbb{R}^d \to \mathbb{R}$, we denote by $\nabla h$ its gradient and $\nabla_i h$ the $i$-th component of the gradient. We use a small constant $\epsilon$, e.g. $10^{-8}$, for numerical stability. Given a sequence $(u_n)_{n \in \mathbb{N}}$ with $\forall n \in \mathbb{N}, u_n \in \mathbb{R}^d$, we denote $u_{n,i}$ for $n \in \mathbb{N}$ and $i \in [d]$ the $i$-th component of the $n$-th element of the sequence.

We want to optimize a function $F : \mathbb{R}^d \to \mathbb{R}$. We assume there exists a random function $f : \mathbb{R}^d \to \mathbb{R}$ such that $\mathbb{E}[\nabla f(x)] = \nabla F(x)$ for all $x \in \mathbb{R}^d$, and that we have access to an oracle providing i.i.d. samples $(f_n)_{n \in \mathbb{N}^*}$. We note $\mathbb{E}_{n-1}[\cdot]$ the conditional expectation knowing $f_1, \ldots, f_{n-1}$. In machine learning, $x$ typically represents the weights of a linear or deep model, $f$ represents the loss from individual training examples or minibatches, and $F$ is the full training objective function. The goal is to find a critical point of $F$.

### 2.2 Adaptive methods

We study both Adagrad (Duchi et al., 2011) and Adam (Kingma & Ba, 2015) using a unified formulation. We assume we have $0 < \beta_2 \leq 1$, $0 \leq \beta_1 < \beta_2$, and a non negative sequence $(\alpha_n)_{n \in \mathbb{N}^*}$. We define three vectors $m_n, v_n, x_n \in \mathbb{R}^d$ iteratively. Given $x_0 \in \mathbb{R}^d$ our starting point, $m_0 = 0$, and $v_0 = 0$, we define for all iterations $n \in \mathbb{N}^*$,

$$m_{n,i} = \beta_1 m_{n-1,i} + \nabla_i f_n(x_{n-1}) \tag{1}$$

$$v_{n,i} = \beta_2 v_{n-1,i} + (\nabla_i f_n(x_{n-1}))^2 \tag{2}$$

$$x_{n,i} = x_{n-1,i} - \alpha_n \frac{m_{n,i}}{\sqrt{\epsilon + v_{n,i}}}. \tag{3}$$

The parameter $\beta_1$ is a heavy-ball style momentum parameter (Polyak, 1964), while $\beta_2$ controls the decay rate of the per-coordinate exponential moving average of the squared gradients. Taking $\beta_1 = 0$, $\beta_2 = 1$ and $\alpha_n = \alpha$ gives Adagrad. While the original Adagrad algorithm did not include a heavy-ball-like momentum, our analysis also applies to the case $\beta_1 > 0$.

**Adam and its corrective terms** The original Adam algorithm (Kingma & Ba, 2015) uses a weighed average, rather than a weighted sum for (1) and (2), i.e. it uses

$$\tilde{m}_{n,i} = (1 - \beta_1) \sum_{k=1}^{n} \beta_1^{n-k} \nabla_i f_n(x_{k-1}) = (1 - \beta_1) m_{n,i},$$

We can achieve the same definition by taking $\alpha_{\text{adam}} = \alpha \cdot \frac{1-\beta_1}{\sqrt{1-\beta_2}}$. The original Adam algorithm further includes two corrective terms to account for the fact that $m_n$ and $v_n$ are biased towards 0 for the first few iterations. Those corrective terms are equivalent to taking a step-size $\alpha_n$ of the form

$$\alpha_{n,\text{adam}} = \alpha \cdot \frac{1-\beta_1}{\sqrt{1-\beta_2}} \cdot \underbrace{\frac{1}{1-\beta_1^n}}_{\substack{\text{corrective} \\ \text{term for } m_n}} \cdot \underbrace{\sqrt{1-\beta_2^n}}_{\substack{\text{corrective} \\ \text{term for } v_n}}. \tag{4}$$

Those corrective terms can be seen as the normalization factors for the weighted sum given by (1) and (2) Note that each term goes to its limit value within a few times $1/(1-\beta)$ updates (with $\beta \in \{\beta_1, \beta_2\}$). which explains the $(1-\beta_1)$ term in (4). In the present work, we propose to drop the corrective term for $m_n$, and to keep only the one for $v_n$, thus using the alternative step size

$$\alpha_n = \alpha(1-\beta_1)\sqrt{\frac{1-\beta_2^n}{1-\beta_2}}. \tag{5}$$

This simplification motivated by several observations:

- By dropping either corrective terms, $\alpha_n$ becomes monotonic, which simplifies the proof.
- For typical values of $\beta_1$ and $\beta_2$ (e.g. 0.9 and 0.999), the corrective term for $m_n$ converges to its limit value much faster than the one for $v_n$.
- Removing the corrective term for $m_n$ is equivalent to a learning-rate warmup, which is popular in deep learning, while removing the one for $v_n$ would lead to an increased step size during early training. For values of $\beta_2$ close to 1, this can lead to divergence in practice.

We experimentally verify in Section 6.3 that dropping the corrective term for $m_n$ has no observable effect on the training process, while dropping the corrective term for $v_n$ leads to observable perturbations. In the following, we thus consider the variation of Adam obtained by taking $\alpha_n$ provided by (5).

## 2.3 Assumptions

We make three assumptions. We first assume $F$ is bounded below by $F_*$, that is,

$$\forall x \in \mathbb{R}^d, \ F(x) \geq F_*. \tag{6}$$

We then assume *the $\ell_\infty$ norm of the stochastic gradients is uniformly almost surely bounded*, i.e. there is $R \geq \sqrt{\epsilon}$ ($\sqrt{\epsilon}$ is used here to simplify the final bounds) so that

$$\forall x \in \mathbb{R}^d, \quad \|\nabla f(x)\|_\infty \leq R - \sqrt{\epsilon} \quad \text{a.s.}, \tag{7}$$

and finally, the *smoothness of the objective function*, e.g., its gradient is $L$-Liptchitz-continuous with respect to the $\ell_2$-norm:

$$\forall x, y \in \mathbb{R}^d, \quad \|\nabla F(x) - \nabla F(y)\|_2 \leq L \|x - y\|_2. \tag{8}$$

We discuss the use of assumption (7) in Section 4.2.

## 3 Related work

Early work on adaptive methods (McMahan & Streeter, 2010; Duchi et al., 2011) showed that Adagrad achieves an optimal rate of convergence of $O(1/\sqrt{N})$ for convex optimization (Agarwal et al., 2009). Later, RMSProp (Tieleman & Hinton, 2012) and Adam (Kingma & Ba, 2015) were developed for training deep neural networks, using an exponential moving average of the past squared gradients.

Kingma & Ba (2015) offered a proof that Adam with a decreasing step size converges for convex objectives. However, the proof contained a mistake spotted by Reddi et al. (2018), who also gave examples of convex

problems where Adam does not converge to an optimal solution. They proposed AMSGrad as a convergent variant, which consisted in retaining the maximum value of the exponential moving average. When $\alpha$ goes to zero, AMSGrad is shown to converge in the convex and non-convex setting (Fang & Klabjan, 2019; Zhou et al., 2018). Despite this apparent flaw in the Adam algorithm, it remains a widely popular optimizer, raising the question as to whether it converges. When $\beta_2$ goes to 1 and $\alpha$ to 0, our results and previous work (Zou et al., 2019a) show that Adam does converge with the same rate as Adagrad. This is coherent with the counter examples of Reddi et al. (2018), because they uses a small exponential decay parameter $\beta_2 < 1/5$.

The convergence of Adagrad for non-convex objectives was first tackled by Li & Orabona (2019), who proved its convergence, but under restrictive conditions (e.g., $\alpha \le \sqrt{\epsilon}/L$). The proof technique was improved by Ward et al. (2019), who showed the convergence of "scalar" Adagrad, i.e., with a single learning rate, for any value of $\alpha$ with a rate of $O(\ln(N)/\sqrt{N})$. Our approach builds on this work but we extend it to both Adagrad and Adam, in their coordinate-wise version, as used in practice, while also supporting heavy-ball momentum.

The coordinate-wise version of Adagrad was also tackled by Zou et al. (2019b), offering a convergence result for Adagrad with either heavy-ball or Nesterov style momentum. We obtain the same rate for heavy-ball momentum with respect to $N$ (i.e., $O(\ln(N)/\sqrt{N})$), but we improve the dependence on the momentum parameter $\beta_1$ from $O((1-\beta_1)^{-3})$ to $O((1-\beta_1)^{-1})$. Chen et al. (2019) also provided a bound for Adagrad and Adam, but without convergence guarantees for Adam for any hyper-parameter choice, and with a worse dependency on $\beta_1$. Zhou et al. (2018) also cover Adagrad in the stochastic setting, however their proof technique leads to a $\sqrt{1/\epsilon}$ term in their bound, typically with $\epsilon = 10^{-8}$. Finally, a convergence bound for Adam was introduced by Zou et al. (2019a). We recover the same scaling of the bound with respect to $\alpha$ and $\beta_2$. However their bound has a dependency of $O((1-\beta_1)^{-5})$ with respect to $\beta_1$, while we get $O((1-\beta_1)^{-1})$, a significant improvement. Shi et al. (2020) obtain similar convergence results for RMSProp and Adam when considering the random shuffling setup. They use an affine growth condition (i.e. norm of the stochastic gradient is bounded by an affine function of the norm of the deterministic gradient) instead of the boundness of the gradient, but their bound decays with the number of total epochs, not stochastic updates leading to an overall $\sqrt{s}$ extra term with $s$ the size of the dataset. Finally, Faw et al. (2022) use the same affine growth assumption to derive high probability bounds for scalar Adagrad.

Non adaptive methods like SGD are also well studied in the non convex setting (Ghadimi & Lan, 2013), with a convergence rate of $O(1/\sqrt{N})$ for a smooth objective with bounded variance of the gradients. Unlike adaptive methods, SGD requires knowing the smoothness constant. When adding heavy-ball momentum, Yang et al. (2016) showed that the convergence bound degrades as $O((1-\beta_1)^{-2})$, assuming that the gradients are bounded. We apply our proof technique for momentum to SGD in the Appendix, Section B and improve this dependency to $O((1-\beta_1)^{-1})$. Recent work by Liu et al. (2020) achieves the same dependency with weaker assumptions. Defazio (2020) provided an in-depth analysis of SGD-M with a tight Liapunov analysis.

## 4 Main results

For a number of iterations $N \in \mathbb{N}^*$, we note $\tau_N$ a random index with value in $\{0, \ldots, N-1\}$, so that

$$\forall j \in \mathbb{N}, j < N, \mathbb{P}\left[\tau = j\right] \propto 1 - \beta_1^{N-j}. \tag{9}$$

If $\beta_1 = 0$, this is equivalent to sampling $\tau$ uniformly in $\{0, \ldots, N-1\}$. If $\beta_1 > 0$, the last few $\frac{1}{1-\beta_1}$ iterations are sampled rarely, and iterations older than a few times that number are sampled almost uniformly. Our results bound the expected squared norm of the gradient at iteration $\tau$, which is standard for non convex stochastic optimization (Ghadimi & Lan, 2013).

### 4.1 Convergence bounds

For simplicity, we first give convergence results for $\beta_1 = 0$, along with a complete proof in Section 5. We then provide the results with momentum, with their proofs in the Appendix, Section A.6. We also provide a bound on the convergence of SGD with a $O(1/(1-\beta_1))$ dependency in the Appendix, Section B.2, along with its proof in Section B.4.

**No heavy-ball momentum**

**Theorem 1** (Convergence of Adagrad without momentum). *Given the assumptions from Section 2.3, the iterates $x_n$ defined in Section 2.2 with hyper-parameters verifying $\beta_2 = 1$, $\alpha_n = \alpha$ with $\alpha > 0$ and $\beta_1 = 0$, and $\tau$ defined by (9), we have for any $N \in \mathbb{N}^*$,*

$$\mathbb{E}\left[\|\nabla F(x_\tau)\|^2\right] \le 2R\frac{F(x_0) - F_*}{\alpha\sqrt{N}} + \frac{1}{\sqrt{N}}\left(4dR^2 + \alpha dRL\right)\ln\left(1 + \frac{NR^2}{\epsilon}\right). \tag{10}$$

**Theorem 2** (Convergence of Adam without momentum). *Given the assumptions from Section 2.3, the iterates $x_n$ defined in Section 2.2 with hyper-parameters verifying $0 < \beta_2 < 1$, $\alpha_n = \alpha\sqrt{\frac{1-\beta_2^n}{1-\beta_2}}$ with $\alpha > 0$ and $\beta_1 = 0$, and $\tau$ defined by (9), we have for any $N \in \mathbb{N}^*$,*

$$\mathbb{E}\left[\|\nabla F(x_\tau)\|^2\right] \le 2R\frac{F(x_0) - F_*}{\alpha N} + E\left(\frac{1}{N}\ln\left(1 + \frac{R^2}{(1-\beta_2)\epsilon}\right) - \ln(\beta_2)\right), \tag{11}$$

*with*

$$E = \frac{4dR^2}{\sqrt{1-\beta_2}} + \frac{\alpha dRL}{1 - \beta_2}.$$

**With heavy-ball momentum**

**Theorem 3** (Convergence of Adagrad with momentum). *Given the assumptions from Section 2.3, the iterates $x_n$ defined in Section 2.2 with hyper-parameters verifying $\beta_2 = 1$, $\alpha_n = \alpha$ with $\alpha > 0$ and $0 \le \beta_1 < 1$, and $\tau$ defined by (9), we have for any $N \in \mathbb{N}^*$ such that $N > \frac{\beta_1}{1-\beta_1}$,*

$$\mathbb{E}\left[\|\nabla F(x_\tau)\|^2\right] \le 2R\sqrt{N}\frac{F(x_0) - F_*}{\alpha\tilde{N}} + \frac{\sqrt{N}}{\tilde{N}}E\ln\left(1 + \frac{NR^2}{\epsilon}\right), \tag{12}$$

*with $\tilde{N} = N - \frac{\beta_1}{1-\beta_1}$, and,*

$$E = \alpha dRL + \frac{12dR^2}{1 - \beta_1} + \frac{2\alpha^2 dL^2\beta_1}{1 - \beta_1}.$$

**Theorem 4** (Convergence of Adam with momentum). *Given the assumptions from Section 2.3, the iterates $x_n$ defined in Section 2.2 with hyper-parameters verifying $0 < \beta_2 < 1$, $0 \le \beta_1 < \beta_2$, and, $\alpha_n = \alpha(1-\beta_1)\sqrt{\frac{1-\beta_2^n}{1-\beta_2}}$ with $\alpha > 0$, and $\tau$ defined by (9), we have for any $N \in \mathbb{N}^*$ such that $N > \frac{\beta_1}{1-\beta_1}$,*

$$\mathbb{E}\left[\|\nabla F(x_\tau)\|^2\right] \le 2R\frac{F(x_0) - F_*}{\alpha\tilde{N}} + E\left(\frac{1}{\tilde{N}}\ln\left(1 + \frac{R^2}{(1-\beta_2)\epsilon}\right) - \frac{N}{\tilde{N}}\ln(\beta_2)\right), \tag{13}$$

*with $\tilde{N} = N - \frac{\beta_1}{1-\beta_1}$, and*

$$E = \frac{\alpha dRL(1-\beta_1)}{(1-\beta_1/\beta_2)(1-\beta_2)} + \frac{12dR^2\sqrt{1-\beta_1}}{(1-\beta_1/\beta_2)^{3/2}\sqrt{1-\beta_2}} + \frac{2\alpha^2 dL^2\beta_1}{(1-\beta_1/\beta_2)(1-\beta_2)^{3/2}}.$$

## 4.2 Analysis of the bounds

**Dependency on $d$.** The dependency in $d$ is present in previous works on coordinate wise adaptive methods (Zou et al., 2019a;b). Note however that $R$ is defined as the $\ell_\infty$ bound on the on the stochastic gradient, so that in the case where the gradient has a similar scale along all dimensions, $dR^2$ would be a reasonable bound for $\|\nabla f(x)\|_2^2$. However, if many dimensions contribute little to the norm of the gradient, this would still lead to a worse dependency in $d$ that e.g. scalar Adagrad Ward et al. (2019) or SGD.

Diving into the technicalities of the proof to come, we will see in Section 5 that we apply Lemma 5.2 once per dimension. The contribution from each coordinate is mostly independent of the actual scale of its gradients (as it only appears in the log), so that the right hand side of the convergence bound will grow as $d$. In contrast, the scalar version of Adagrad (Ward et al., 2019) has a single learning rate, so that Lemma 5.2 is only applied once, removing the dependency on $d$. However, this variant is rarely used in practice.

**Almost sure bound on the gradient.** We chose to assume the existence of an almost sure uniform $\ell_\infty$-bound on the gradients given by (7). This is a strong assumption, although it is weaker than the one used by Duchi et al. (2011) for Adagrad in the convex case, where the iterates were assumed to be almost surely bounded. There exist a few real life problems that verifies this assumption, for instance logistic regression without weight penalty, and with bounded inputs. It is possible instead to assume only a uniform bound on the expected gradient $\nabla F(x)$, as done by Ward et al. (2019) and Zou et al. (2019b). This however lead to a bound on $\mathbb{E}\left[\|\nabla F(x_\tau)\|_2^{4/3}\right]^{2/3}$ instead of a bound on $\mathbb{E}\left[\|\nabla F(x_\tau)\|_2^2\right]$, all the other terms staying the same. We provide the sketch of the proof using Hölder inequality in the Appendix, Section A.7.

It is also possible to replace the bound on the gradient with an affine growth condition, i.e. the norm of the stochastic gradient is bounded by an affine function of the norm of the expected gradient. A proof for scalar Adagrad is provided by Faw et al. (2022). Shi et al. (2020) do the same for RMSProp, however their convergence bound is decays as $O(\log(T)/\sqrt{T})$ with $T$ the number of epoch, not the number of updates, leading to a significantly less tight bound for large datasets.

**Impact of heavy-ball momentum.** Looking at Theorems 3 and 4, we see that increasing $\beta_1$ always deteriorates the bounds. Taking $\beta_1 = 0$ in those theorems gives us almost exactly the bound without heavy-ball momentum from Theorems 1 and 2, up to a factor 3 in the terms of the form $dR^2$.

As discussed in Section 3, previous bounds for Adagrad in the non-convex setting deteriorates as $O((1-\beta_1)^{-3})$ (Zou et al., 2019b), while bounds for Adam deteriorates as $O((1-\beta_1)^{-5})$ (Zou et al., 2019a). Our unified proof for Adam and Adagrad achieves a dependency of $O((1-\beta_1)^{-1})$, a significant improvement. We refer the reader to the Appendix, Section A.3, for a detailed analysis. While our dependency still contradicts the benefits of using momentum observed in practice, see Section 6, our tighter analysis is a step in the right direction.

**On sampling of $\tau$** Note that in (9), we sample with a lower probability the latest iterations. This can be explained by the fact that the proof technique for stochastic optimization in the non-convex case is based on the idea that for every iteration $n$, either $\nabla F(x_n)$ is small, or $F(x_{n+1})$ will decrease by some amount. However, when introducing momentum, and especially when taking the limit $\beta_1 \to 1$, the latest gradient $\nabla F(x_n)$ has almost no influence over $x_{n+1}$, as the momentum term updates slowly. Momentum *spreads* the influence of the gradients over time, and thus, it will take a few updates for a gradient to have fully influenced the iterate $x_n$ and thus the value of the function $F(x_n)$. From a formal point of view, the sampling weights given by (9) naturally appear as part of the proof which is presented in Section A.6.

### 4.3 Optimal finite horizon Adam is Adagrad

Let us take a closer look at the result from Theorem 2. It could seem like some quantities can explode but actually not for any reasonable values of $\alpha$, $\beta_2$ and $N$. Let us try to find the best possible rate of convergence for Adam for a finite horizon $N$, i.e. $q \in \mathbb{R}_+$ such that $\mathbb{E}\left[\|\nabla F(x_\tau)\|^2\right] = O(\ln(N)N^{-q})$ for some choice of the hyper-parameters $\alpha(N)$ and $\beta_2(N)$. Given that the upper bound in (11) is a sum of non-negative terms, we need each term to be of the order of $\ln(N)N^{-q}$ or negligible. Let us assume that this rate is achieved for $\alpha(N)$ and $\beta_2(N)$. The bound tells us that convergence can only be achieved if $\lim \alpha(N) = 0$ and $\lim \beta_2(N) = 1$, with the limits taken for $N \to \infty$. This motivates us to assume that there exists an asymptotic development of $\alpha(N) \propto N^{-a} + o(N^{-a})$, and of $1 - \beta_2(N) \propto N^{-b} + o(N^{-b})$ for $a$ and $b$ positive. Thus, let us consider only the leading term in those developments, ignoring the leading constant (which is assumed to be non-zero). Let us further assume that $\epsilon \ll R^2$, we have

$$\mathbb{E}\left[\|\nabla F(x_\tau)\|^2\right] \leq 2R\frac{F(x_0) - F_*}{N^{1-a}} + E\left(\frac{1}{N}\ln\left(\frac{R^2 N^b}{\epsilon}\right) + \frac{N^{-b}}{1 - N^{-b}}\right), \tag{14}$$

with $E = 4dR^2N^{b/2} + dRLN^{b-a}$. Let us ignore the log terms for now, and use $\frac{N^{-b}}{1-N^{-b}} \sim N^{-b}$ for $N \to \infty$ , to get

$$\mathbb{E}\left[\|\nabla F(x_\tau)\|^2\right] \lesssim 2R\frac{F(x_0) - F_*}{N^{1-a}} + 4dR^2N^{b/2-1} + 4dR^2N^{-b/2} + dRLN^{b-a-1} + dRLN^{-a}.$$

Adding back the logarithmic term, the best rate we can obtain is $O(\ln(N)/\sqrt{N})$, and it is only achieved for $a = 1/2$ and $b = 1$, i.e., $\alpha = \alpha_1/\sqrt{N}$ and $\beta_2 = 1 - 1/N$. We can see the resemblance between Adagrad on one side and Adam with a finite horizon and such parameters on the other. Indeed, an exponential moving average with a parameter $\beta_2 = 1 - 1/N$ as a typical averaging window length of size $N$, while Adagrad would be an exact average of the past $N$ terms. In particular, the bound for Adam now becomes

$$\mathbb{E}\left[\|\nabla F(x_\tau)\|^2\right] \leq \frac{F(x_0) - F_*}{\alpha_1\sqrt{N}} + \frac{1}{\sqrt{N}}\left(4dR^2 + \alpha_1 dRL\right)\left(\ln\left(1 + \frac{RN}{\epsilon}\right) + \frac{N}{N-1}\right), \tag{15}$$

which differ from (10) only by a $+N/(N-1)$ next to the log term.

**Adam and Adagrad are twins.** Our analysis highlights an important fact: *Adam is to Adagrad like constant step size SGD is to decaying step size SGD.* While Adagrad is asymptotically optimal, it also leads to a slower decrease of the term proportional to $F(x_0) - F_*$, as $1/\sqrt{N}$ instead of $1/N$ for Adam. During the initial phase of training, it is likely that this term dominates the loss, which could explain the popularity of Adam for training deep neural networks rather than Adagrad. With its default parameters, Adam will not converge. It is however possible to choose $\alpha$ and $\beta_2$ to achieve an $\epsilon$ critical point for $\epsilon$ arbitrarily small and, for a known time horizon, they can be chosen to obtain the exact same bound as Adagrad.

## 5 Proofs for $\beta_1 = 0$ (no momentum)

We assume here for simplicity that $\beta_1 = 0$, i.e., there is no heavy-ball style momentum. Taking $n \in \mathbb{N}^*$, the recursions introduced in Section 2.2 can be simplified into

$$\begin{cases} v_{n,i} &= \beta_2 v_{n-1,i} + \left(\nabla_i f_n(x_{n-1})\right)^2, \\ x_{n,i} &= x_{n-1,i} - \alpha_n \frac{\nabla_i f_n(x_{n-1})}{\sqrt{\epsilon + v_{n,i}}}. \end{cases} \tag{16}$$

Remember that we recover Adagrad when $\alpha_n = \alpha$ for $\alpha > 0$ and $\beta_2 = 1$, while Adam can be obtained taking $0 < \beta_2 < 1$, $\alpha > 0$,

$$\alpha_n = \alpha\sqrt{\frac{1 - \beta_2^n}{1 - \beta_2}}, \tag{17}$$

Throughout the proof we denote by $\mathbb{E}_{n-1}[\cdot]$ the conditional expectation with respect to $f_1, \ldots, f_{n-1}$. In particular, $x_{n-1}$ and $v_{n-1}$ are deterministic knowing $f_1, \ldots, f_{n-1}$. For all $n \in \mathbb{N}^*$, we also define $\tilde{v}_n \in \mathbb{R}^d$ so that for all $i \in [d]$,

$$\tilde{v}_{n,i} = \beta_2 v_{n-1,i} + \mathbb{E}_{n-1}\left[\left(\nabla_i f_n(x_{n-1})\right)^2\right], \tag{18}$$

i.e., we replace the last gradient contribution by its expected value conditioned on $f_1, \ldots, f_{n-1}$.

### 5.1 Technical lemmas

A problem posed by the update (16) is the correlation between the numerator and denominator. This prevents us from easily computing the conditional expectation and as noted by Reddi et al. (2018), the expected direction of update can have a positive dot product with the objective gradient. It is however possible to control the deviation from the descent direction, following Ward et al. (2019) with this first lemma.

**Lemma 5.1** (adaptive update approximately follow a descent direction)**.** *For all $n \in \mathbb{N}^*$ and $i \in [d]$, we have:*

$$\mathbb{E}_{n-1}\left[\nabla_i F(x_{n-1})\frac{\nabla_i f_n(x_{n-1})}{\sqrt{\epsilon + v_{n,i}}}\right] \geq \frac{(\nabla_i F(x_{n-1}))^2}{2\sqrt{\epsilon + \tilde{v}_{n,i}}} - 2R\mathbb{E}_{n-1}\left[\frac{(\nabla_i f_n(x_{n-1}))^2}{\epsilon + v_{n,i}}\right]. \tag{19}$$

*Proof.* We take $i \in [d]$ and note $G = \nabla_i F(x_{n-1})$, $g = \nabla_i f_n(x_{n-1})$, $v = v_{n,i}$ and $\tilde{v} = \tilde{v}_{n,i}$.

$$\mathbb{E}_{n-1}\left[\frac{Gg}{\sqrt{\epsilon+v}}\right] = \mathbb{E}_{n-1}\left[\frac{Gg}{\sqrt{\epsilon+\tilde{v}}}\right] + \mathbb{E}_{n-1}\left[\underbrace{Gg\left(\frac{1}{\sqrt{\epsilon+v}} - \frac{1}{\sqrt{\epsilon+\tilde{v}}}\right)}_{A}\right]. \tag{20}$$

Given that $g$ and $\tilde{v}$ are independent knowing $f_1, \ldots, f_{n-1}$, we immediately have

$$\mathbb{E}_{n-1}\left[\frac{Gg}{\sqrt{\epsilon+\tilde{v}}}\right] = \frac{G^2}{\sqrt{\epsilon+\tilde{v}}}. \tag{21}$$

Now we need to control the size of the second term $A$,

$$A = Gg\frac{\tilde{v}-v}{\sqrt{\epsilon+v}\sqrt{\epsilon+\tilde{v}}(\sqrt{\epsilon+v}+\sqrt{\epsilon+\tilde{v}})},$$

$$= Gg\frac{\mathbb{E}_{n-1}\left[g^2\right]-g^2}{\sqrt{\epsilon+v}\sqrt{\epsilon+\tilde{v}}(\sqrt{\epsilon+v}+\sqrt{\epsilon+\tilde{v}})}$$

$$|A| \leq \underbrace{|Gg|\frac{\mathbb{E}_{n-1}\left[g^2\right]}{\sqrt{\epsilon+v}(\epsilon+\tilde{v})}}_{\kappa} + \underbrace{|Gg|\frac{g^2}{(\epsilon+v)\sqrt{\epsilon+\tilde{v}}}}_{\rho}.$$

The last inequality comes from the fact that $\sqrt{\epsilon+v}+\sqrt{\epsilon+\tilde{v}} \geq \max(\sqrt{\epsilon+v}, \sqrt{\epsilon+\tilde{v}})$ and $\left|\mathbb{E}_{n-1}\left[g^2\right]-g^2\right| \leq \mathbb{E}_{n-1}\left[g^2\right]+g^2$. Following Ward et al. (2019), we can use the following inequality to bound $\kappa$ and $\rho$,

$$\forall \lambda > 0, \, x, y \in \mathbb{R}, xy \leq \frac{\lambda}{2}x^2 + \frac{y^2}{2\lambda}. \tag{22}$$

First applying (22) to $\kappa$ with

$$\lambda = \frac{\sqrt{\epsilon+\tilde{v}}}{2}, \, x = \frac{|G|}{\sqrt{\epsilon+\tilde{v}}}, \, y = \frac{|g|\,\mathbb{E}_{n-1}\left[g^2\right]}{\sqrt{\epsilon+\tilde{v}}\sqrt{\epsilon+v}},$$

we obtain

$$\kappa \leq \frac{G^2}{4\sqrt{\epsilon+\tilde{v}}} + \frac{g^2\mathbb{E}_{n-1}\left[g^2\right]^2}{(\epsilon+\tilde{v})^{3/2}(\epsilon+v)}.$$

Given that $\epsilon + \tilde{v} \geq \mathbb{E}_{n-1}\left[g^2\right]$ and taking the conditional expectation, we can simplify as

$$\mathbb{E}_{n-1}\left[\kappa\right] \leq \frac{G^2}{4\sqrt{\epsilon+\tilde{v}}} + \frac{\mathbb{E}_{n-1}\left[g^2\right]}{\sqrt{\epsilon+\tilde{v}}}\mathbb{E}_{n-1}\left[\frac{g^2}{\epsilon+v}\right]. \tag{23}$$

Given that $\sqrt{\mathbb{E}_{n-1}\left[g^2\right]} \leq \sqrt{\epsilon+\tilde{v}}$ and $\sqrt{\mathbb{E}_{n-1}\left[g^2\right]} \leq R$, we can simplify (23) as

$$\mathbb{E}_{n-1}\left[\kappa\right] \leq \frac{G^2}{4\sqrt{\epsilon+\tilde{v}}} + R\mathbb{E}_{n-1}\left[\frac{g^2}{\epsilon+v}\right]. \tag{24}$$

Now turning to $\rho$, we use (22) with

$$\lambda = \frac{\sqrt{\epsilon+\tilde{v}}}{2\mathbb{E}_{n-1}\left[g^2\right]}, \, x = \frac{|Gg|}{\sqrt{\epsilon+\tilde{v}}}, \, y = \frac{g^2}{\epsilon+v},$$

we obtain

$$\rho \leq \frac{G^2}{4\sqrt{\epsilon+\tilde{v}}}\frac{g^2}{\mathbb{E}_{n-1}\left[g^2\right]} + \frac{\mathbb{E}_{n-1}\left[g^2\right]}{\sqrt{\epsilon+\tilde{v}}}\frac{g^4}{(\epsilon+v)^2}, \tag{25}$$

Given that $\epsilon + v \geq g^2$ and taking the conditional expectation we obtain

$$\mathbb{E}_{n-1}[\rho] \leq \frac{G^2}{4\sqrt{\epsilon + \tilde{v}}} + \frac{\mathbb{E}_{n-1}[g^2]}{\sqrt{\epsilon + \tilde{v}}} \mathbb{E}_{n-1}\left[\frac{g^2}{\epsilon + v}\right], \tag{26}$$

which we simplify using the same argument as for (24) into

$$\mathbb{E}_{n-1}[\rho] \leq \frac{G^2}{4\sqrt{\epsilon + \tilde{v}}} + R\mathbb{E}_{n-1}\left[\frac{g^2}{\epsilon + v}\right]. \tag{27}$$

Notice that in (25), we possibly divide by zero. It suffice to notice that if $\mathbb{E}_{n-1}[g^2] = 0$ then $g^2 = 0$ a.s. so that $\rho = 0$ and (27) is still verified. Summing (24) and (27) we can bound

$$\mathbb{E}_{n-1}[|A|] \leq \frac{G^2}{2\sqrt{\epsilon + \tilde{v}}} + 2R\mathbb{E}_{n-1}\left[\frac{g^2}{\epsilon + v}\right]. \tag{28}$$

Injecting (28) and (21) into (20) finishes the proof. $\qquad\square$

Anticipating on Section 5.2, the previous Lemma gives us a bound on the deviation from a descent direction. While for a specific iteration, this deviation can take us away from a descent direction, the next lemma tells us that the sum of those deviations cannot grow larger than a logarithmic term. This key insight introduced in Ward et al. (2019) is what makes the proof work.

**Lemma 5.2** (sum of ratios with the denominator being the sum of past numerators). *We assume we have $0 < \beta_2 \leq 1$ and a non-negative sequence $(a_n)_{n \in \mathbb{N}^*}$. We define for all $n \in \mathbb{N}^*$, $b_n = \sum_{j=1}^n \beta_2^{n-j} a_j$. We have*

$$\sum_{j=1}^N \frac{a_j}{\epsilon + b_j} \leq \ln\left(1 + \frac{b_N}{\epsilon}\right) - N\ln(\beta_2). \tag{29}$$

*Proof.* Given that ln is increasing, and the fact that $b_j > a_j \geq 0$, we have for all $j \in \mathbb{N}^*$,

$$\frac{a_j}{\epsilon + b_j} \leq \ln(\epsilon + b_j) - \ln(\epsilon + b_j - a_j)$$

$$= \ln(\epsilon + b_j) - \ln(\epsilon + \beta_2 b_{j-1})$$

$$= \ln\left(\frac{\epsilon + b_j}{\epsilon + b_{j-1}}\right) + \ln\left(\frac{\epsilon + b_{j-1}}{\epsilon + \beta_2 b_{j-1}}\right).$$

The first term forms a telescoping series, while the second one is bounded by $-\ln(\beta_2)$. Summing over all $j \in [N]$ gives the desired result. $\qquad\square$

## 5.2 Proof of Adam and Adagrad without momentum

Let us take an iteration $n \in \mathbb{N}^*$, we define the update $u_n \in \mathbb{R}^d$:

$$\forall i \in [d], u_{n,i} = \frac{\nabla_i f_n(x_{n-1})}{\sqrt{\epsilon + v_{n,i}}}. \tag{30}$$

**Adagrad.** As explained in Section 2.2, we have $\alpha_n = \alpha$ for $\alpha > 0$. Using the smoothness of $F$ (8), we have

$$F(x_{n+1}) \leq F(x_n) - \alpha\nabla F(x_n)^T u_n + \frac{\alpha^2 L}{2}\|u_n\|_2^2. \tag{31}$$

Taking the conditional expectation with respect to $f_0, \ldots, f_{n-1}$ we can apply the descent Lemma 5.1. Notice that due to the a.s. $\ell_\infty$ bound on the gradients (7), we have for any $i \in [d]$, $\sqrt{\epsilon + \tilde{v}_{n,i}} \leq R\sqrt{n}$, so that,

$$\frac{\alpha\left(\nabla_i F(x_{n-1})\right)^2}{2\sqrt{\epsilon + \tilde{v}_{n,i}}} \geq \frac{\alpha\left(\nabla_i F(x_{n-1})\right)^2}{2R\sqrt{n}}. \tag{32}$$

This gives us

$$\mathbb{E}_{n-1}\left[F(x_n)\right] \leq F(x_{n-1}) - \frac{\alpha}{2R\sqrt{n}}\left\|\nabla F(x_{n-1})\right\|_2^2 + \left(2\alpha R + \frac{\alpha^2 L}{2}\right)\mathbb{E}_{n-1}\left[\left\|u_n\right\|_2^2\right].$$

Summing the previous inequality for all $n \in [N]$, taking the complete expectation, and using that $\sqrt{n} \leq \sqrt{N}$ gives us,

$$\mathbb{E}\left[F(x_N)\right] \leq F(x_0) - \frac{\alpha}{2R\sqrt{N}}\sum_{n=0}^{N-1}\mathbb{E}\left[\left\|\nabla F(x_n)\right\|_2^2\right] + \left(2\alpha R + \frac{\alpha^2 L}{2}\right)\sum_{n=0}^{N-1}\mathbb{E}\left[\left\|u_n\right\|_2^2\right].$$

From there, we can bound the last sum on the right hand side using Lemma 5.2 once for each dimension. Rearranging the terms, we obtain the result of Theorem 1.

**Adam.**  As given by (5) in Section 2.2, we have $\alpha_n = \alpha\sqrt{\frac{1-\beta_2^n}{1-\beta_2}}$ for $\alpha > 0$. Using the smoothness of $F$ defined in (8), we have

$$F(x_n) \leq F(x_{n-1}) - \alpha_n \nabla F(x_{n-1})^T u_n + \frac{\alpha_n^2 L}{2}\left\|u_n\right\|_2^2. \tag{33}$$

We have for any $i \in [d]$, $\sqrt{\epsilon + \tilde{v}_{n,i}} \leq R\sqrt{\sum_{j=0}^{n-1}\beta_2^j} = R\sqrt{\frac{1-\beta_2^n}{1-\beta_2}}$, thanks to the a.s. $\ell_\infty$ bound on the gradients (7), so that,

$$\alpha_n \frac{\left(\nabla_i F(x_{n-1})\right)^2}{2\sqrt{\epsilon + \tilde{v}_{n,i}}} \geq \frac{\alpha\left(\nabla_i F(x_{n-1})\right)^2}{2R}. \tag{34}$$

Taking the conditional expectation with respect to $f_1, \ldots, f_{n-1}$ we can apply the descent Lemma 5.1 and use (34) to obtain from (33),

$$\mathbb{E}_{n-1}\left[F(x_n)\right] \leq F(x_{n-1}) - \frac{\alpha}{2R}\left\|\nabla F(x_{n-1})\right\|_2^2 + \left(2\alpha_n R + \frac{\alpha_n^2 L}{2}\right)\mathbb{E}_{n-1}\left[\left\|u_n\right\|_2^2\right].$$

Given that $\beta_2 < 1$, we have $\alpha_n \leq \frac{\alpha}{\sqrt{1-\beta_2}}$. Summing the previous inequality for all $n \in [N]$ and taking the complete expectation yields

$$\mathbb{E}\left[F(x_N)\right] \leq F(x_0) - \frac{\alpha}{2R}\sum_{n=0}^{N-1}\mathbb{E}\left[\left\|\nabla F(x_n)\right\|_2^2\right] + \left(\frac{2\alpha R}{\sqrt{1-\beta_2}} + \frac{\alpha^2 L}{2(1-\beta_2)}\right)\sum_{n=0}^{N-1}\mathbb{E}\left[\left\|u_n\right\|_2^2\right].$$

Applying Lemma 5.2 for each dimension and rearranging the terms finishes the proof of Theorem 2.

## 6  Experiments

On Figure 1, we compare the effective dependency of the average squared norm of the gradient in the parameters $\alpha$, $\beta_1$ and $\beta_2$ for Adam, when used on a toy task and CIFAR-10.

### 6.1  Setup

**Toy problem.**  In order to support the bounds presented in Section 4, in particular the dependency in $\beta_2$, we test Adam on a specifically crafted toy problem. We take $x \in \mathbb{R}^6$ and define for all $i \in [6]$, $p_i = 10^{-i}$. We take $(Q_i)_{i \in [6]}$, Bernoulli variables with $\mathbb{P}[Q_i = 1] = p_i$. We then define $f$ for all $x \in \mathbb{R}^d$ as

$$f(x) = \sum_{i \in [6]}(1 - Q_i)\,\mathrm{Huber}(x_i - 1) + \frac{Q_i}{\sqrt{p_i}}\,\mathrm{Huber}(x_i + 1), \tag{35}$$

with for all $y \in \mathbb{R}$,

$$\mathrm{Huber}(y) = \begin{cases} \frac{y^2}{2} & \text{when } |y| \leq 1 \\ |y| - \frac{1}{2} & \text{otherwise.} \end{cases}$$

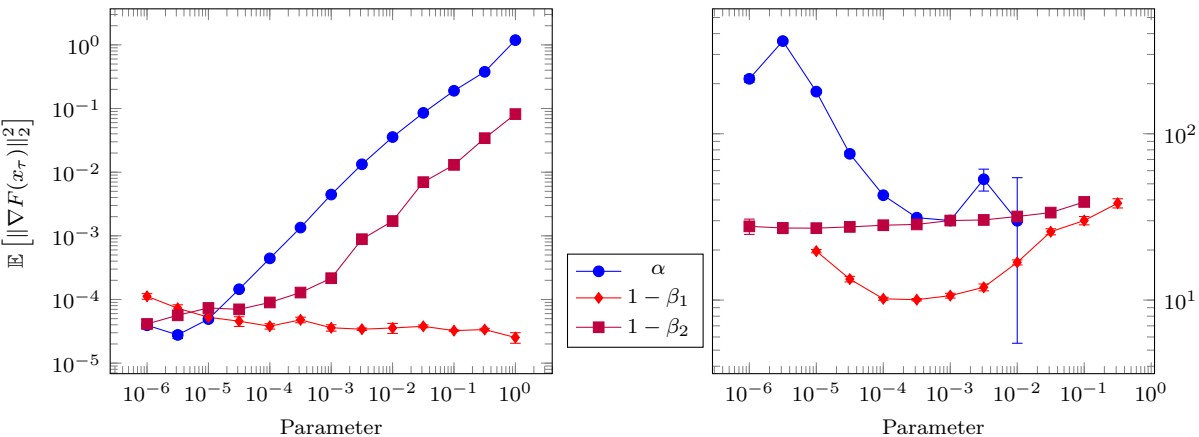

(a) Average squared norm of the gradient on a toy task, see Section 6, for more details. For the $\alpha$ and $1 - \beta_2$ curves, we initialize close to the optimum to make the $F_0 - F_*$ term negligible.

(b) Average squared norm of the gradient of a small convolutional model Gitman & Ginsburg (2017) trained on CIFAR-10, with a random initialization. The full gradient is evaluated every epoch.

Figure 1: Observed average squared norm of the objective gradients after a fixed number of iterations when varying a single parameter out of $\alpha$, $1 - \beta_1$ and $1 - \beta_2$, on a toy task (left, $10^6$ iterations) and on CIFAR-10 (right, 600 epochs with a batch size 128). All curves are averaged over 3 runs, error bars are negligible except for small values of $\alpha$ on CIFAR-10. See Section 6 for details.

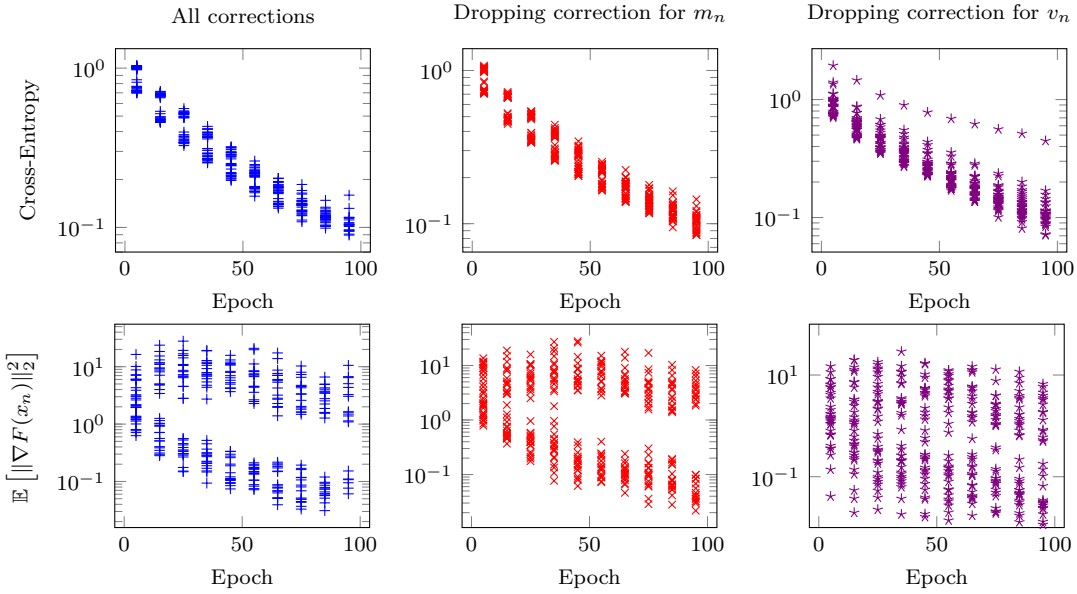

Figure 2: Training trajectories for varying values of $\alpha \in \{10^{-4}, 10^{-3}\}$, $\beta_1 \in \{0., 0.5, 0.8, 0.9, 0.99\}$ and $\beta_2 \in \{0.9, 0.99, 0.999, 0.9999\}$. The top row (resp. bottom) gives the training loss (resp. squared norm of the expected gradient). The left column uses all corrective terms in the original Adam algorithm, the middle column drops the corrective term on $m_n$ (equivalent to our proof setup), and the right column drops the corrective term on $v_n$. We notice a limited impact when dropping the corrective term on $m_n$, but dropping the corrective term on $v_n$ has a much stronger impact.

Intuitively, each coordinate is pointing most of the time towards 1, but exceptionally towards -1 with a weight of $1/\sqrt{p_i}$. Those rare events happens less and less often as $i$ increase, but with an increasing weight.

Those weights are chosen so that all the coordinates of the gradient have the same variance[1]. It is necessary to take different probabilities for each coordinate. If we use the same $p$ for all, we observe a phase transition when $1 - \beta_2 \approx p$, but not the continuous improvement we obtain on Figure 1a.

We plot the variation of $\mathbb{E}\left[\|F(x_\tau)\|_2^2\right]$ after $10^6$ iterations with batch size 1 when varying either $\alpha$, $1 - \beta_1$ or $1 - \beta_2$ through a range of 13 values uniformly spaced in log-scale between $10^{-6}$ and 1. When varying $\alpha$, we take $\beta_1 = 0$ and $\beta_2 = 1 - 10^{-6}$. When varying $\beta_1$, we take $\alpha = 10^{-5}$ and $\beta_2 = 1 - 10^{-6}$ (i.e. $\beta_2$ is so that we are in the Adagrad-like regime). Finally, when varying $\beta_2$, we take $\beta_1 = 0$ and $\alpha = 10^{-6}$. We start from $x_0$ close to the optimum by running first $10^6$ iterations with $\alpha = 10^{-4}$, then $10^6$ iterations with $\alpha = 10^{-5}$, always with $\beta_2 = 1 - 10^{-6}$. This allows to have $F(x_0) - F_* \approx 0$ in (11) and (13) and focus on the second part of both bounds. All curves are averaged over three runs. Error bars are plotted but not visible in log-log scale.

**CIFAR-10.** We train a simple convolutional network (Gitman & Ginsburg, 2017) on the CIFAR-10[2] image classification dataset. Starting from a random initialization, we train the model on a single V100 for 600 epochs with a batch size of 128, evaluating the full training gradient after each epoch. This is a proxy for $\mathbb{E}\left[\|F(x_\tau)\|_2^2\right]$, which would be to costly to evaluate exactly. All runs use the default config $\alpha = 10^{-3}$, $\beta_2 = 0.999$ and $\beta_1 = 0.9$, and we then change one of the parameter.

We take $\alpha$ from a uniform range in log-space between $10^{-6}$ and $10^{-2}$ with 9 values, for $1 - \beta_1$ the range is from $10^{-5}$ to 0.3 with 9 values, and for $1 - \beta_2$, from $10^{-6}$ to $10^{-1}$ with 11 values. Unlike for the toy problem, we do not initialize close to the optimum, as even after 600 epochs, the norm of the gradients indicates that we are not at a critical point. All curves are averaged over three runs. Error bars are plotted but not visible in log-log scale, except for large values of $\alpha$.

## 6.2 Analysis

**Toy problem.** Looking at Figure 1a, we observe a continual improvement as $\beta_2$ increases. Fitting a linear regression in log-log scale of $\mathbb{E}[\|\nabla F(x_\tau)\|_2^2]$ with respect to $1 - \beta_2$ gives a slope of 0.56 which is compatible with our bound (11), in particular the dependency in $O(1/\sqrt{1 - \beta_2})$. As we initialize close to the optimum, a small step size $\alpha$ yields as expected the best performance. Doing the same regression in log-log scale, we find a slope of 0.87, which is again compatible with the $O(\alpha)$ dependency of the second term in (11). Finally, we observe a limited impact of $\beta_1$, except when $1 - \beta_1$ is small. The regression in log-log scale gives a slope of -0.16, while our bound predicts a slope of -1.

**CIFAR 10.** Let us now turn to Figure 1b. As we start from random weights for this problem, we observe that a large step size gives the best performance, although we observe a high variance for the largest $\alpha$. This indicates that training becomes unstable for large $\alpha$, which is not predicted by the theory. This is likely a consequence of the bounded gradient assumption (7) not being verified for deep neural networks. We observe a small improvement as $1 - \beta_2$ decreases, although nowhere near what we observed on our toy problem. Finally, we observe a sweet spot for the momentum $\beta_1$, not predicted by our theory. We conjecture that this is due to the variance reduction effect of momentum (averaging of the gradients over multiple mini-batches, while the weights have not moved so much as to invalidate past information).

## 6.3 Impact of the Adam corrective terms

Using the same experimental setup on CIFAR-10, we compare the impact of removing either of the corrective term of the original Adam algorithm (Kingma & Ba, 2015), as discussed in Section 2.2. We ran a cartesian product of training for 100 epochs, with $\beta_1 \in \{0, 0.5, 0.8, 0.9, 0.99\}$, $\beta_2 \in \{0.9, 0.99, 0.999, 0.9999\}$, and $\alpha \in \{10^{-4}, 10^{-3}\}$. We report both the training loss and norm of the expected gradient on Figure 2. We notice a limited difference when dropping the corrective term on $m_n$, but dropping the term $v_n$ has an

---

[1] We deviate from the a.s. bounded gradient assumption for this experiment, see Section 4.2 for a discussion on a.s. bound vs bound in expectation.

[2] https://www.cs.toronto.edu/~kriz/cifar.html

important impact on the training trajectories. This confirm our motivation for simplifying the proof by removing the corrective term on the momentum.

## 7 Conclusion

We provide a simple proof on the convergence of Adam and Adagrad without heavy-ball style momentum. Our analysis highlights a link between the two algorithms: with right the hyper-parameters, Adam converges like Adagrad. The extension to heavy-ball momentum is more complex, but we significantly improve the dependence on the momentum parameter for Adam, Adagrad, as well as SGD. We exhibit a toy problem where the dependency on $\alpha$ and $\beta_2$ experimentally matches our prediction. However, we do not predict the practical interest of momentum, so that improvements to the proof are needed for future work.

### Broader Impact Statement

The present theoretical results on the optimization of non convex losses in a stochastic settings impact our understanding of the training of deep neural network. It might allow a deeper understanding of neural network training dynamics and thus reinforce any existing deep learning applications. There would be however no direct possible negative impact to society.

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

# Supplementary material for A Simple Convergence Proof of Adam and Adagrad

## Overview

In Section A, we detail the results for the convergence of Adam and Adagrad with heavy-ball momentum. For an overview of the contributions of our proof technique, see Section A.4.

Then in Section B, we show how our technique also applies to SGD and improves its dependency in $\beta_1$ compared with previous work by Yang et al. (2016), from $O((1-\beta_1)^{-2})$ to $O(1-\beta_1)^{-1}$. The proof is simpler than for Adam/Adagrad, and show the generality of our technique.

## A    Convergence of adaptive methods with heavy-ball momentum

### A.1    Setup and notations

We recall the dynamic system introduced in Section 2.3. In the rest of this section, we take an iteration $n \in \mathbb{N}^*$, and when needed, $i \in [d]$ refers to a specific coordinate. Given $x_0 \in \mathbb{R}^d$ our starting point, $m_0 = 0$, and $v_0 = 0$, we define

$$
\begin{cases}
m_{n,i} & = \beta_1 m_{n-1,i} + \nabla_i f_n(x_{n-1}), \\
v_{n,i} & = \beta_2 v_{n-1,i} + \left( \nabla_i f_n(x_{n-1}) \right)^2, \\
x_{n,i} & = x_{n-1,i} - \alpha_n \frac{m_{n,i}}{\sqrt{\epsilon + v_{n,i}}}.
\end{cases}
\tag{A.1}
$$

For Adam, the step size is given by

$$
\alpha_n = \alpha(1 - \beta_1)\sqrt{\frac{1 - \beta_2^n}{1 - \beta_2}}.
\tag{A.2}
$$

For Adagrad (potentially extended with heavy-ball momentum), we have $\beta_2 = 1$ and

$$
\alpha_n = \alpha(1 - \beta_1).
\tag{A.3}
$$

Notice we include the factor $1 - \beta_1$ in the step size rather than in (A.1), as this allows for a more elegant proof. The original Adam algorithm included compensation factors for both $\beta_1$ and $\beta_2$ (Kingma & Ba, 2015) to correct the initial scale of $m$ and $v$ which are initialized at 0. Adam would be exactly recovered by replacing (A.2) with

$$
\alpha_n = \alpha \frac{1 - \beta_1}{1 - \beta_1^n}\sqrt{\frac{1 - \beta_2^n}{1 - \beta_2}}.
\tag{A.4}
$$

However, the denominator $1 - \beta_1^n$ potentially makes $(\alpha_n)_{n \in \mathbb{N}^*}$ non monotonic, which complicates the proof. Thus, we instead replace the denominator by its limit value for $n \to \infty$. This has little practical impact as (i) early iterates are noisy because $v$ is averaged over a small number of gradients, so making smaller step can be more stable, (ii) for $\beta_1 = 0.9$ (Kingma & Ba, 2015), (A.2) differs from (A.4) only for the first 50 iterations.

Throughout the proof we note $\mathbb{E}_{n-1}[\cdot]$ the conditional expectation with respect to $f_1, \dots, f_{n-1}$. In particular, $x_{n-1}, v_{n-1}$ is deterministic knowing $f_1, \dots, f_{n-1}$. We introduce

$$
G_n = \nabla F(x_{n-1}) \quad \text{and} \quad g_n = \nabla f_n(x_{n-1}).
\tag{A.5}
$$

Like in Section 5.2, we introduce the update $u_n \in \mathbb{R}^d$, as well as the update without heavy-ball momentum $U_n \in \mathbb{R}^d$:

$$
u_{n,i} = \frac{m_{n,i}}{\sqrt{\epsilon + v_{n,i}}} \quad \text{and} \quad U_{n,i} = \frac{g_{n,i}}{\sqrt{\epsilon + v_{n,i}}}.
\tag{A.6}
$$

For any $k \in \mathbb{N}$ with $k < n$, we define $\tilde{v}_{n,k} \in \mathbb{R}^d$ by

$$\tilde{v}_{n,k,i} = \beta_2^k v_{n-k,i} + \mathbb{E}_{n-k-1}\left[\sum_{j=n-k+1}^{n} \beta_2^{n-j} g_{j,i}^2\right], \tag{A.7}$$

i.e. the contribution from the $k$ last gradients are replaced by their expected value for know values of $f_1, \ldots, f_{n-k-1}$. For $k = 1$, we recover the same definition as in (18).

## A.2 Results

For any total number of iterations $N \in \mathbb{N}^*$, we define $\tau_N$ a random index with value in $\{0, \ldots, N-1\}$, verifying

$$\forall j \in \mathbb{N}, j < N, \mathbb{P}\left[\tau = j\right] \propto 1 - \beta_1^{N-j}. \tag{A.8}$$

If $\beta_1 = 0$, this is equivalent to sampling $\tau$ uniformly in $\{0, \ldots, N-1\}$. If $\beta_1 > 0$, the last few $\frac{1}{1-\beta_1}$ iterations are sampled rarely, and all iterations older than a few times that number are sampled almost uniformly. We bound the expected squared norm of the total gradient at iteration $\tau$, which is standard for non convex stochastic optimization (Ghadimi & Lan, 2013).

Note that like in previous works, the bound worsen as $\beta_1$ increases, with a dependency of the form $O((1 - \beta_1)^{-1})$. This is a significant improvement over the existing bound for Adagrad with heavy-ball momentum, which scales as $(1-\beta_1)^{-3}$ (Zou et al., 2019b), or the best known bound for Adam which scales as $(1-\beta_1)^{-5}$ (Zou et al., 2019a).

Technical lemmas to prove the following theorems are introduced in Section A.5, while the proof of Theorems 3 and 4 are provided in Section A.6.

**Theorem 3** (Convergence of Adagrad with momentum)**.** *Given the assumptions from Section 2.3, the iterates $x_n$ defined in Section 2.2 with hyper-parameters verifying $\beta_2 = 1$, $\alpha_n = \alpha$ with $\alpha > 0$ and $0 \le \beta_1 < 1$, and $\tau$ defined by (9), we have for any $N \in \mathbb{N}^*$ such that $N > \frac{\beta_1}{1-\beta_1}$,*

$$\mathbb{E}\left[\|\nabla F(x_\tau)\|^2\right] \le 2R\sqrt{N}\frac{F(x_0) - F_*}{\alpha\tilde{N}} + \frac{\sqrt{N}}{\tilde{N}}E\ln\left(1 + \frac{NR^2}{\epsilon}\right), \tag{12}$$

*with $\tilde{N} = N - \frac{\beta_1}{1-\beta_1}$, and,*

$$E = \alpha dRL + \frac{12dR^2}{1-\beta_1} + \frac{2\alpha^2 dL^2\beta_1}{1-\beta_1}.$$

**Theorem 4** (Convergence of Adam with momentum)**.** *Given the assumptions from Section 2.3, the iterates $x_n$ defined in Section 2.2 with hyper-parameters verifying $0 < \beta_2 < 1$, $0 \le \beta_1 < \beta_2$, and, $\alpha_n = \alpha(1 - \beta_1)\sqrt{\frac{1-\beta_2^n}{1-\beta_2}}$ with $\alpha > 0$, and $\tau$ defined by (9), we have for any $N \in \mathbb{N}^*$ such that $N > \frac{\beta_1}{1-\beta_1}$,*

$$\mathbb{E}\left[\|\nabla F(x_\tau)\|^2\right] \le 2R\frac{F(x_0) - F_*}{\alpha\tilde{N}} + E\left(\frac{1}{\tilde{N}}\ln\left(1 + \frac{R^2}{(1-\beta_2)\epsilon}\right) - \frac{N}{\tilde{N}}\ln(\beta_2)\right), \tag{13}$$

*with $\tilde{N} = N - \frac{\beta_1}{1-\beta_1}$, and*

$$E = \frac{\alpha dRL(1-\beta_1)}{(1-\beta_1/\beta_2)(1-\beta_2)} + \frac{12dR^2\sqrt{1-\beta_1}}{(1-\beta_1/\beta_2)^{3/2}\sqrt{1-\beta_2}} + \frac{2\alpha^2 dL^2\beta_1}{(1-\beta_1/\beta_2)(1-\beta_2)^{3/2}}.$$

## A.3 Analysis of the results with momentum

First notice that taking $\beta_1 \to 0$ in Theorems 3 and 4, we almost recover the same result as stated in 2 and 1, only losing on the term $4dR^2$ which becomes $12dR^2$.

**Simplified expressions with momentum** Assuming $N \gg \frac{\beta_1}{1-\beta_1}$ and $\beta_1/\beta_2 \approx \beta_1$, which is verified for typical values of $\beta_1$ and $\beta_2$ (Kingma & Ba, 2015), it is possible to simplify the bound for Adam (13) as

$$
\mathbb{E}\left[\|\nabla F(x_\tau)\|^2\right] \lesssim 2R\frac{F(x_0) - F_*}{\alpha N}
$$
$$
+ \left(\frac{\alpha dRL}{1-\beta_2} + \frac{12dR^2}{(1-\beta_1)\sqrt{1-\beta_2}} + \frac{2\alpha^2 dL^2\beta_1}{(1-\beta_1)(1-\beta_2)^{3/2}}\right)\left(\frac{1}{N}\ln\left(1 + \frac{R^2}{\epsilon(1-\beta_2)}\right) - \ln(\beta_2)\right). \quad \text{(A.9)}
$$

Similarly, if we assume $N \gg \frac{\beta_1}{1-\beta_1}$, we can simplify the bound for Adagrad (**??**) as

$$
\mathbb{E}\left[\|\nabla F(x_\tau)\|^2\right] \lesssim 2R\frac{F(x_0) - F_*}{\alpha\sqrt{N}} + \frac{1}{\sqrt{N}}\left(\alpha dRL + \frac{12dR^2}{1-\beta_1} + \frac{2\alpha^2 dL^2\beta_1}{1-\beta_1}\right)\ln\left(1 + \frac{NR^2}{\epsilon}\right), \quad \text{(A.10)}
$$

**Optimal finite horizon Adam is still Adagrad** We can perform the same finite horizon analysis as in Section 4.3. If we take $\alpha = \frac{\tilde{\alpha}}{\sqrt{N}}$ and $\beta_2 = 1 - 1/N$, then (A.9) simplifies to

$$
\mathbb{E}\left[\|\nabla F(x_\tau)\|^2\right] \lesssim 2R\frac{F(x_0) - F_*}{\tilde{\alpha}\sqrt{N}} + \frac{1}{\sqrt{N}}\left(\tilde{\alpha}dRL + \frac{12dR^2}{1-\beta_1} + \frac{2\tilde{\alpha}^2 dL^2\beta_1}{1-\beta_1}\right)\left(\ln\left(1 + \frac{NR^2}{\epsilon}\right) + 1\right). \quad \text{(A.11)}
$$

The term $(1-\beta_2)^{3/2}$ in the denominator in (A.9) is indeed compensated by the $\alpha^2$ in the numerator and we again recover the proper $\ln(N)/\sqrt{N}$ convergence rate, which matches (A.10) up to a $+1$ term next to the log.

## A.4 Overview of the proof, contributions and limitations

There is a number of steps to the proof. First we derive a Lemma similar in spirit to the descent Lemma 5.1. There are two differences: first, when computing the dot product between the current expected gradient and each past gradient contained in the momentum, we have to re-center the expected gradient to its values in the past, using the smoothness assumption. Besides, we now have to decorrelate more terms between the numerator and denominator, as the numerator contains not only the latest gradient but a decaying sum of the past ones. We similarly extend Lemma 5.2 to support momentum specific terms. The rest of the proof follows mostly as in Section 5, except with a few more manipulation to regroup the gradient terms coming from different iterations.

Compared with previous work (Zou et al., 2019b;a), the re-centering of past gradients in (A.14) is a key aspect to improve the dependency in $\beta_1$, with a small price to pay using the smoothness of $F$ which is compensated by the introduction of extra $G^2_{n-k,i}$ in (A.1). Then, a tight handling of the different summations as well as the the introduction of a non uniform sampling of the iterates (A.8), which naturally arises when grouping the different terms in (A.49), allow to obtain the overall improved dependency in $O((1-\beta_1)^{-1})$.

The same technique can be applied to SGD, the proof becoming simpler as there is no correlation between the step size and the gradient estimate, see Section B. If you want to better understand the handling of momentum without the added complexity of adaptive methods, we recommend starting with this proof.

A limitation of the proof technique is that we do not show that heavy-ball momentum can lead to a variance reduction of the update. Either more powerful probabilistic results, or extra regularity assumptions could allow to further improve our worst case bounds of the variance of the update, which in turn might lead to a bound with an improvement when using heavy-ball momentum.

## A.5 Technical lemmas

We first need an updated version of 5.1 that includes momentum.

**Lemma A.1** (Adaptive update with momentum approximately follows a descent direction). *Given $x_0 \in \mathbb{R}^d$, the iterates defined by the system* (A.1) *for $(\alpha_j)_{j \in \mathbb{N}^*}$ that is non-decreasing, and under the conditions* (6),

(7), *and* (8), *as well as* $0 \leq \beta_1 < \beta_2 \leq 1$, *we have for all iterations* $n \in \mathbb{N}^*$,

$$
\mathbb{E}\left[\sum_{i \in [d]} G_{n,i} \frac{m_{n,i}}{\sqrt{\epsilon + v_{n,i}}}\right] \geq \frac{1}{2}\left(\sum_{i \in [d]} \sum_{k=0}^{n-1} \beta_1^k \mathbb{E}\left[\frac{G_{n-k,i}^2}{\sqrt{\epsilon + \tilde{v}_{n,k+1,i}}}\right]\right)
$$
$$
- \frac{\alpha_n^2 L^2}{4R}\sqrt{1 - \beta_1}\left(\sum_{l=1}^{n-1}\|u_{n-l}\|_2^2 \sum_{k=l}^{n-1} \beta_1^k \sqrt{k}\right) - \frac{3R}{\sqrt{1 - \beta_1}}\left(\sum_{k=0}^{n-1}\left(\frac{\beta_1}{\beta_2}\right)^k \sqrt{k+1}\,\|U_{n-k}\|_2^2\right). \quad \text{(A.12)}
$$

*Proof.* We use multiple times (22) in this proof, which we repeat here for convenience,

$$
\forall \lambda > 0,\, x, y \in \mathbb{R},\, xy \leq \frac{\lambda}{2}x^2 + \frac{y^2}{2\lambda}. \quad \text{(A.13)}
$$

Let us take an iteration $n \in \mathbb{N}^*$ for the duration of the proof. We have

$$
\sum_{i \in [d]} G_{n,i} \frac{m_{n,i}}{\sqrt{\epsilon + v_{n,i}}} = \sum_{i \in [d]} \sum_{k=0}^{n-1} \beta_1^k G_{n,i} \frac{g_{n-k,i}}{\sqrt{\epsilon + v_{n,i}}}
$$
$$
= \underbrace{\sum_{i \in [d]} \sum_{k=0}^{n-1} \beta_1^k G_{n-k,i} \frac{g_{n-k,i}}{\sqrt{\epsilon + v_{n,i}}}}_{A} + \underbrace{\sum_{i \in [d]} \sum_{k=0}^{n-1} \beta_1^k \left(G_{n,i} - G_{n-k,i}\right) \frac{g_{n-k,i}}{\sqrt{\epsilon + v_{n,i}}}}_{B}, \quad \text{(A.14)}
$$

Let us now take an index $0 \leq k \leq n-1$. We show that the contribution of past gradients $G_{n-k}$ and $g_{n-k}$ due to the heavy-ball momentum can be controlled thanks to the decay term $\beta_1^k$. Let us first have a look at $B$. Using (A.13) with

$$
\lambda = \frac{\sqrt{1 - \beta_1}}{2R\sqrt{k+1}},\; x = |G_{n,i} - G_{n-k,i}|,\; y = \frac{|g_{n-k,i}|}{\sqrt{\epsilon + v_{n,i}}},
$$

we have

$$
|B| \leq \sum_{i \in [d]} \sum_{k=0}^{n-1} \beta_1^k \left(\frac{\sqrt{1 - \beta_1}}{4R\sqrt{k+1}}\left(G_{n,i} - G_{n-k,i}\right)^2 + \frac{R\sqrt{k+1}}{\sqrt{1 - \beta_1}} \frac{g_{n-k,i}^2}{\epsilon + v_{n,i}}\right). \quad \text{(A.15)}
$$

Notice first that for any dimension $i \in [d]$, $\epsilon + v_{n,i} \geq \epsilon + \beta_2^k v_{n-k,i} \geq \beta_2^k(\epsilon + v_{n-k,i})$, so that

$$
\frac{g_{n-k,i}^2}{\epsilon + v_{n,i}} \leq \frac{1}{\beta_2^k} U_{n-k,i}^2 \quad \text{(A.16)}
$$

Besides, using the L-smoothness of $F$ given by (8), we have

$$
\|G_n - G_{n-k}\|_2^2 \leq L^2 \|x_{n-1} - x_{n-k-1}\|_2^2
$$
$$
= L^2 \left\|\sum_{l=1}^{k} \alpha_{n-l} u_{n-l}\right\|_2^2
$$
$$
\leq \alpha_n^2 L^2 k \sum_{l=1}^{k} \|u_{n-l}\|_2^2, \quad \text{(A.17)}
$$

using Jensen inequality and the fact that $\alpha_n$ is non-decreasing. Injecting (A.16) and (A.17) into (A.15), we obtain

$$
|B| \leq \left(\sum_{k=0}^{n-1} \frac{\alpha_n^2 L^2}{4R}\sqrt{1 - \beta_1}\beta_1^k \sqrt{k} \sum_{l=1}^{k}\|u_{n-l}\|_2^2\right) + \left(\sum_{k=0}^{n-1} \frac{R}{\sqrt{1 - \beta_1}}\left(\frac{\beta_1}{\beta_2}\right)^k \sqrt{k+1}\,\|U_{n-k}\|_2^2\right)
$$
$$
= \sqrt{1 - \beta_1}\frac{\alpha_n^2 L^2}{4R}\left(\sum_{l=1}^{n-1}\|u_{n-l}\|_2^2 \sum_{k=l}^{n-1} \beta_1^k \sqrt{k}\right) + \frac{R}{\sqrt{1 - \beta_1}}\left(\sum_{k=0}^{n-1}\left(\frac{\beta_1}{\beta_2}\right)^k \sqrt{k+1}\,\|U_{n-k}\|_2^2\right). \quad \text{(A.18)}
$$

Now going back to the $A$ term in (A.14), we will study the main term of the summation, i.e. for $i \in [d]$ and $k < n$

$$\mathbb{E}\left[G_{n-k,i}\frac{g_{n-k,i}}{\sqrt{\epsilon + v_{n,i}}}\right] = \mathbb{E}\left[\nabla_i F(x_{n-k-1})\frac{\nabla_i f_{n-k}(x_{n-k-1})}{\sqrt{\epsilon + v_{n,i}}}\right]. \tag{A.19}$$

Notice that we could almost apply Lemma 5.1 to it, except that we have $v_{n,i}$ in the denominator instead of $v_{n-k,i}$. Thus we will need to extend the proof to decorrelate more terms. We will further drop indices in the rest of the proof, noting $G = G_{n-k,i}$, $g = g_{n-k,i}$, $\tilde{v} = \tilde{v}_{n,k+1,i}$ and $v = v_{n,i}$. Finally, let us note

$$\delta^2 = \sum_{j=n-k}^{n}\beta_2^{n-j}g_{j,i}^2 \qquad \text{and} \qquad r^2 = \mathbb{E}_{n-k-1}\left[\delta^2\right]. \tag{A.20}$$

In particular we have $\tilde{v} - v = r^2 - \delta^2$. With our new notations, we can rewrite (A.19) as

$$
\begin{aligned}
\mathbb{E}\left[G\frac{g}{\sqrt{\epsilon + v}}\right] &= \mathbb{E}\left[G\frac{g}{\sqrt{\epsilon + \tilde{v}}} + Gg\left(\frac{1}{\sqrt{\epsilon + v}} - \frac{1}{\sqrt{\epsilon + \tilde{v}}}\right)\right] \\
&= \mathbb{E}\left[\mathbb{E}_{n-k-1}\left[G\frac{g}{\sqrt{\epsilon + \tilde{v}}}\right] + Gg\frac{r^2 - \delta^2}{\sqrt{\epsilon + v}\sqrt{\epsilon + \tilde{v}}(\sqrt{\epsilon + v} + \sqrt{\epsilon + \tilde{v}})}\right] \\
&= \mathbb{E}\left[\frac{G^2}{\sqrt{\epsilon + \tilde{v}}}\right] + \mathbb{E}\Big[\underbrace{Gg\frac{r^2 - \delta^2}{\sqrt{\epsilon + v}\sqrt{\epsilon + \tilde{v}}(\sqrt{\epsilon + v} + \sqrt{\epsilon + \tilde{v}})}}_{C}\Big].
\end{aligned}
\tag{A.21}
$$

We first focus on $C$:

$$|C| \leq \underbrace{|Gg|\frac{r^2}{\sqrt{\epsilon + v}(\epsilon + \tilde{v})}}_{\kappa} + \underbrace{|Gg|\frac{\delta^2}{(\epsilon + v)\sqrt{\epsilon + \tilde{v}}}}_{\rho},$$

due to the fact that $\sqrt{\epsilon + v} + \sqrt{\epsilon + \tilde{v}} \geq \max(\sqrt{\epsilon + v}, \sqrt{\epsilon + \tilde{v}})$ and $\left|r^2 - \delta^2\right| \leq r^2 + \delta^2$.

Applying (A.13) to $\kappa$ with

$$\lambda = \frac{\sqrt{1 - \beta_1}\sqrt{\epsilon + \tilde{v}}}{2}, \quad x = \frac{|G|}{\sqrt{\epsilon + \tilde{v}}}, \quad y = \frac{|g|\, r^2}{\sqrt{\epsilon + \tilde{v}}\sqrt{\epsilon + v}},$$

we obtain

$$\kappa \leq \frac{G^2}{4\sqrt{\epsilon + \tilde{v}}} + \frac{1}{\sqrt{1 - \beta_1}}\frac{g^2 r^4}{(\epsilon + \tilde{v})^{3/2}(\epsilon + v)}.$$

Given that $\epsilon + \tilde{v} \geq r^2$ and taking the conditional expectation, we can simplify as

$$\mathbb{E}_{n-k-1}\left[\kappa\right] \leq \frac{G^2}{4\sqrt{\epsilon + \tilde{v}}} + \frac{1}{\sqrt{1 - \beta_1}}\frac{r^2}{\sqrt{\epsilon + \tilde{v}}}\mathbb{E}_{n-k-1}\left[\frac{g^2}{\epsilon + v}\right]. \tag{A.22}$$

Now turning to $\rho$, we use (A.13) with

$$\lambda = \frac{\sqrt{1 - \beta_1}\sqrt{\epsilon + \tilde{v}}}{2r^2}, \quad x = \frac{|G\delta|}{\sqrt{\epsilon + \tilde{v}}}, \quad y = \frac{|\delta g|}{\epsilon + v},$$

we obtain

$$\rho \leq \frac{G^2}{4\sqrt{\epsilon + \tilde{v}}}\frac{\delta^2}{r^2} + \frac{1}{\sqrt{1 - \beta_1}}\frac{r^2}{\sqrt{\epsilon + \tilde{v}}}\frac{g^2\delta^2}{(\epsilon + v)^2}. \tag{A.23}$$

Given that $\epsilon + v \geq \delta^2$, and $\mathbb{E}_{n-k-1}\left[\frac{\delta^2}{r^2}\right] = 1$, we obtain after taking the conditional expectation,

$$\mathbb{E}_{n-k-1}\left[\rho\right] \leq \frac{G^2}{4\sqrt{\epsilon+\tilde{v}}} + \frac{1}{\sqrt{1-\beta_1}}\frac{r^2}{\sqrt{\epsilon+\tilde{v}}}\mathbb{E}_{n-k-1}\left[\frac{g^2}{\epsilon+v}\right]. \tag{A.24}$$

Notice that in A.23, we possibly divide by zero. It suffice to notice that if $r^2 = 0$ then $\delta^2 = 0$ a.s. so that $\rho = 0$ and (A.24) is still verified. Summing (A.22) and (A.24), we get

$$\mathbb{E}_{n-k-1}\left[|C|\right] \leq \frac{G^2}{2\sqrt{\epsilon+\tilde{v}}} + \frac{2}{\sqrt{1-\beta_1}}\frac{r^2}{\sqrt{\epsilon+\tilde{v}}}\mathbb{E}_{n-k-1}\left[\frac{g^2}{\epsilon+v}\right]. \tag{A.25}$$

Given that $r \leq \sqrt{\epsilon+\tilde{v}}$ by definition of $\tilde{v}$, and that using (7), $r \leq \sqrt{k+1}R$, we have[3], reintroducing the indices we had dropped

$$\mathbb{E}_{n-k-1}\left[|C|\right] \leq \frac{G_{n-k,i}^2}{2\sqrt{\epsilon+\tilde{v}_{n,k+1,i}}} + \frac{2R}{\sqrt{1-\beta_1}}\sqrt{k+1}\mathbb{E}_{n-k-1}\left[\frac{g_{n-k,i}^2}{\epsilon+v_{n,i}}\right]. \tag{A.26}$$

Taking the complete expectation and using that by definition $\epsilon + v_{n,i} \geq \epsilon + \beta_2^k v_{n-k,i} \geq \beta_2^k(\epsilon+v_{n-k,i})$ we get

$$\mathbb{E}\left[|C|\right] \leq \frac{1}{2}\mathbb{E}\left[\frac{G_{n-k,i}^2}{\sqrt{\epsilon+\tilde{v}_{n,k+1,i}}}\right] + \frac{2R}{\sqrt{1-\beta_1}\beta_2^k}\sqrt{k+1}\mathbb{E}\left[\frac{g_{n-k,i}^2}{\epsilon+v_{n-k,i}}\right]. \tag{A.27}$$

Injecting (A.27) into (A.21) gives us

$$\mathbb{E}\left[A\right] \geq \sum_{i\in[d]}\sum_{k=0}^{n-1}\beta_1^k\left(\mathbb{E}\left[\frac{G_{n-k,i}^2}{\sqrt{\epsilon+\tilde{v}_{n,k+1,i}}}\right] - \left(\frac{1}{2}\mathbb{E}\left[\frac{G_{n-k,i}^2}{\sqrt{\epsilon+\tilde{v}_{n,k,i}}}\right] + \frac{2R}{\sqrt{1-\beta_1}\beta_2^k}\sqrt{k+1}\mathbb{E}\left[\frac{g_{n-k,i}^2}{\epsilon+v_{n-k,i}}\right]\right)\right)$$

$$= \frac{1}{2}\left(\sum_{i\in[d]}\sum_{k=0}^{n-1}\beta_1^k\mathbb{E}\left[\frac{G_{n-k,i}^2}{\sqrt{\epsilon+\tilde{v}_{n,k+1,i}}}\right]\right) - \frac{2R}{\sqrt{1-\beta_1}}\left(\sum_{i\in[d]}\sum_{k=0}^{n-1}\left(\frac{\beta_1}{\beta_2}\right)^k\sqrt{k+1}\mathbb{E}\left[\|U_{n-k}\|_2^2\right]\right). \tag{A.28}$$

Injecting (A.28) and (A.18) into (A.14) finishes the proof.

$\square$

Similarly, we will need an updated version of 5.2.

**Lemma A.2** (sum of ratios of the square of a decayed sum and a decayed sum of square). *We assume we have $0 < \beta_2 \leq 1$ and $0 < \beta_1 < \beta_2$, and a sequence of real numbers $(a_n)_{n\in\mathbb{N}^*}$. We define $b_n = \sum_{j=1}^n \beta_2^{n-j}a_j^2$ and $c_n = \sum_{j=1}^n \beta_1^{n-j}a_j$. Then we have*

$$\sum_{j=1}^n \frac{c_j^2}{\epsilon+b_j} \leq \frac{1}{(1-\beta_1)(1-\beta_1/\beta_2)}\left(\ln\left(1+\frac{b_n}{\epsilon}\right) - n\ln(\beta_2)\right). \tag{A.29}$$

*Proof.* Now let us take $j \in \mathbb{N}^*$, $j \leq n$, we have using Jensen inequality

$$c_j^2 \leq \frac{1}{1-\beta_1}\sum_{l=1}^j \beta_1^{j-l}a_l^2,$$

so that

$$\frac{c_j^2}{\epsilon+b_j} \leq \frac{1}{1-\beta_1}\sum_{l=1}^j \beta_1^{j-l}\frac{a_l^2}{\epsilon+b_j}.$$

---

[3]Note that we do not need the almost sure bound on the gradient, and a bound on $\mathbb{E}\left[\|\nabla f(x)\|_\infty^2\right]$ would be sufficient.

Given that for $l \in [j]$, we have by definition $\epsilon + b_j \geq \epsilon + \beta_2^{j-l} b_l \geq \beta_2^{j-l}(\epsilon + b_l)$, we get

$$\frac{c_j^2}{\epsilon + b_j} \leq \frac{1}{1 - \beta_1} \sum_{l=1}^{j} \left(\frac{\beta_1}{\beta_2}\right)^{j-l} \frac{a_l^2}{\epsilon + b_l}. \tag{A.30}$$

Thus, when summing over all $j \in [n]$, we get

$$\sum_{j=1}^{n} \frac{c_j^2}{\epsilon + b_j} \leq \frac{1}{1 - \beta_1} \sum_{j=1}^{n} \sum_{l=1}^{j} \left(\frac{\beta_1}{\beta_2}\right)^{j-l} \frac{a_l^2}{\epsilon + b_l}$$

$$= \frac{1}{1 - \beta_1} \sum_{l=1}^{n} \frac{a_l^2}{\epsilon + b_l} \sum_{j=l}^{n} \left(\frac{\beta_1}{\beta_2}\right)^{j-l}$$

$$\leq \frac{1}{(1 - \beta_1)(1 - \beta_1/\beta_2)} \sum_{l=1}^{n} \frac{a_l^2}{\epsilon + b_l}. \tag{A.31}$$

Applying Lemma 5.2, we obtain (A.29).

$\square$

We also need two technical lemmas on the sum of series.

**Lemma A.3** (sum of a geometric term times a square root). *Given $0 < a < 1$ and $Q \in \mathbb{N}$, we have,*

$$\sum_{q=0}^{Q-1} a^q \sqrt{q+1} \leq \frac{1}{1-a} \left(1 + \frac{\sqrt{\pi}}{2\sqrt{-\ln(a)}}\right) \leq \frac{2}{(1-a)^{3/2}}. \tag{A.32}$$

*Proof.* We first need to study the following integral:

$$\int_0^\infty \frac{a^x}{2\sqrt{x}} \mathrm{d}x = \int_0^\infty \frac{\mathrm{e}^{\ln(a)x}}{2\sqrt{x}} \mathrm{d}x \quad, \text{ then introducing } y = \sqrt{x},$$

$$= \int_0^\infty \mathrm{e}^{\ln(a)y^2} \mathrm{d}y \quad, \text{ then introducing } u = \sqrt{-2\ln(a)}y,$$

$$= \frac{1}{\sqrt{-2\ln(a)}} \int_0^\infty \mathrm{e}^{-u^2/2} \mathrm{d}u$$

$$\int_0^\infty \frac{a^x}{2\sqrt{x}} \mathrm{d}x = \frac{\sqrt{\pi}}{2\sqrt{-\ln(a)}}, \tag{A.33}$$

where we used the classical integral of the standard Gaussian density function.

Let us now introduce $A_Q$:

$$A_Q = \sum_{q=0}^{Q-1} a^q \sqrt{q+1},$$

then we have

$$A_Q - aA_Q = \sum_{q=0}^{Q-1} a^q \sqrt{q+1} - \sum_{q=1}^{Q} a^q \sqrt{q} \quad, \text{ then using the concavity of } \sqrt{\cdot},$$

$$\leq 1 - a^Q \sqrt{Q} + \sum_{q=1}^{Q-1} \frac{a^q}{2\sqrt{q}}$$

$$\leq 1 + \int_0^\infty \frac{a^x}{2\sqrt{x}} \mathrm{d}x$$

$$(1-a)A_Q \leq 1 + \frac{\sqrt{\pi}}{2\sqrt{-\ln(a)}},$$

where we used (A.33). Given that $\sqrt{-\ln(a)} \geq \sqrt{1-a}$ we obtain (A.32). $\qquad\square$

**Lemma A.4** (sum of a geometric term times roughly a power 3/2)**.** *Given $0 < a < 1$ and $Q \in \mathbb{N}$, we have,*

$$\sum_{q=0}^{Q-1} a^q \sqrt{q}(q+1) \leq \frac{4a}{(1-a)^{5/2}}. \tag{A.34}$$

*Proof.* Let us introduce $A_Q$:

$$A_Q = \sum_{q=0}^{Q-1} a^q \sqrt{q}(q+1),$$

then we have

$$
\begin{aligned}
A_Q - aA_Q &= \sum_{q=0}^{Q-1} a^q \sqrt{q}(q+1) - \sum_{q=1}^{Q} a^q \sqrt{q-1}\,q \\
&\leq \sum_{q=1}^{Q-1} a^q \sqrt{q}\left((q+1) - \sqrt{q}\sqrt{q-1}\right) \\
&\leq \sum_{q=1}^{Q-1} a^q \sqrt{q}\left((q+1) - (q-1)\right) \\
&\leq 2 \sum_{q=1}^{Q-1} a^q \sqrt{q} \\
&= 2a \sum_{q=0}^{Q-2} a^q \sqrt{q+1} \quad \text{, then using Lemma A.3,} \\
(1-a)A_Q &\leq \frac{4a}{(1-a)^{3/2}}.
\end{aligned}
$$

$\qquad\square$

### A.6 Proof of Adam and Adagrad with momentum

**Common part of the proof** Let us a take an iteration $n \in \mathbb{N}^*$. Using the smoothness of $F$ defined in (8), we have

$$F(x_n) \leq F(x_{n-1}) - \alpha_n G_n^T u_n + \frac{\alpha_n^2 L}{2} \|u_n\|_2^2.$$

Taking the full expectation and using Lemma A.1,

$$
\begin{aligned}
\mathbb{E}\left[F(x_n)\right] \leq{}& \mathbb{E}\left[F(x_{n-1})\right] - \frac{\alpha_n}{2}\left(\sum_{i \in [d]}\sum_{k=0}^{n-1}\beta_1^k \mathbb{E}\left[\frac{G_{n-k,i}^2}{2\sqrt{\epsilon + \tilde{v}_{n,k+1,i}}}\right]\right) + \frac{\alpha_n^2 L}{2}\mathbb{E}\left[\|u_n\|_2^2\right] \\
&+ \frac{\alpha_n^3 L^2}{4R}\sqrt{1-\beta_1}\left(\sum_{l=1}^{n-1}\|u_{n-l}\|_2^2 \sum_{k=l}^{n-1}\beta_1^k\sqrt{k}\right) + \frac{3\alpha_n R}{\sqrt{1-\beta_1}}\left(\sum_{k=0}^{n-1}\left(\frac{\beta_1}{\beta_2}\right)^k\sqrt{k+1}\,\|U_{n-k}\|_2^2\right). \tag{A.35}
\end{aligned}
$$

Notice that because of the bound on the $\ell_\infty$ norm of the stochastic gradients at the iterates (7), we have for any $k \in \mathbb{N}$, $k < n$, and any coordinate $i \in [d]$, $\sqrt{\epsilon + \tilde{v}_{n,k+1,i}} \leq R\sqrt{\sum_{j=0}^{n-1}\beta_2^j}$. Introducing $\Omega_n = \sqrt{\sum_{j=0}^{n-1}\beta_2^j}$,

we have

$$\mathbb{E}\left[F(x_n)\right] \le \mathbb{E}\left[F(x_{n-1})\right] - \frac{\alpha_n}{2R\Omega_n} \sum_{k=0}^{n-1} \beta_1^k \mathbb{E}\left[\|G_{n-k}\|_2^2\right] + \frac{\alpha_n^2 L}{2} \mathbb{E}\left[\|u_n\|_2^2\right]$$

$$+ \frac{\alpha_n^3 L^2}{4R} \sqrt{1-\beta_1} \left(\sum_{l=1}^{n-1} \|u_{n-l}\|_2^2 \sum_{k=l}^{n-1} \beta_1^k \sqrt{k}\right) + \frac{3\alpha_n R}{\sqrt{1-\beta_1}} \left(\sum_{k=0}^{n-1} \left(\frac{\beta_1}{\beta_2}\right)^k \sqrt{k+1} \|U_{n-k}\|_2^2\right). \quad (A.36)$$

Now summing over all iterations $n \in [N]$ for $N \in \mathbb{N}^*$, and using that for both Adam (A.2) and Adagrad (A.3), $\alpha_n$ is non-decreasing, as well the fact that $F$ is bounded below by $F_*$ from (6), we get

$$\underbrace{\frac{1}{2R} \sum_{n=1}^{N} \frac{\alpha_n}{\Omega_n} \sum_{k=0}^{n-1} \beta_1^k \mathbb{E}\left[\|G_{n-k}\|_2^2\right]}_{A} \le F(x_0) - F_* + \underbrace{\frac{\alpha_N^2 L}{2} \sum_{n=1}^{N} \mathbb{E}\left[\|u_n\|_2^2\right]}_{B}$$

$$+ \underbrace{\frac{\alpha_N^3 L^2}{4R} \sqrt{1-\beta_1} \sum_{n=1}^{N} \sum_{l=1}^{n-1} \mathbb{E}\left[\|u_{n-l}\|_2^2\right] \sum_{k=l}^{n-1} \beta_1^k \sqrt{k}}_{C} + \underbrace{\frac{3\alpha_N R}{\sqrt{1-\beta_1}} \sum_{n=1}^{N} \sum_{k=0}^{n-1} \left(\frac{\beta_1}{\beta_2}\right)^k \sqrt{k+1} \mathbb{E}\left[\|U_{n-k}\|_2^2\right]}_{D}. \quad (A.37)$$

First looking at $B$, we have using Lemma A.2,

$$B \le \frac{\alpha_N^2 L}{2(1-\beta_1)(1-\beta_1/\beta_2)} \sum_{i\in[d]} \left(\ln\left(1 + \frac{v_{N,i}}{\epsilon}\right) - N\log(\beta_2)\right). \quad (A.38)$$

Then looking at $C$ and introducing the change of index $j = n - l$,

$$C = \frac{\alpha_N^3 L^2}{4R} \sqrt{1-\beta_1} \sum_{n=1}^{N} \sum_{j=1}^{n} \mathbb{E}\left[\|u_j\|_2^2\right] \sum_{k=n-j}^{n-1} \beta_1^k \sqrt{k}$$

$$= \frac{\alpha_N^3 L^2}{4R} \sqrt{1-\beta_1} \sum_{j=1}^{N} \mathbb{E}\left[\|u_j\|_2^2\right] \sum_{n=j}^{N} \sum_{k=n-j}^{n-1} \beta_1^k \sqrt{k}$$

$$= \frac{\alpha_N^3 L^2}{4R} \sqrt{1-\beta_1} \sum_{j=1}^{N} \mathbb{E}\left[\|u_j\|_2^2\right] \sum_{k=0}^{N-1} \beta_1^k \sqrt{k} \sum_{n=j}^{j+k} 1$$

$$= \frac{\alpha_N^3 L^2}{4R} \sqrt{1-\beta_1} \sum_{j=1}^{N} \mathbb{E}\left[\|u_j\|_2^2\right] \sum_{k=0}^{N-1} \beta_1^k \sqrt{k}(k+1)$$

$$\le \frac{\alpha_N^3 L^2}{R} \sum_{j=1}^{N} \mathbb{E}\left[\|u_j\|_2^2\right] \frac{\beta_1}{(1-\beta_1)^2}, \quad (A.39)$$

using Lemma A.4. Finally, using Lemma A.2, we get

$$C \le \frac{\alpha_N^3 L^2 \beta_1}{R(1-\beta_1)^3(1-\beta_1/\beta_2)} \sum_{i\in[d]} \left(\ln\left(1 + \frac{v_{N,i}}{\epsilon}\right) - N\log(\beta_2)\right). \quad (A.40)$$

Finally, introducing the same change of index $j = n - k$ for $D$, we get

$$D = \frac{3\alpha_N R}{\sqrt{1 - \beta_1}} \sum_{n=1}^{N} \sum_{j=1}^{n} \left(\frac{\beta_1}{\beta_2}\right)^{n-j} \sqrt{1 + n - j} \mathbb{E}\left[\|U_j\|_2^2\right]$$

$$= \frac{3\alpha_N R}{\sqrt{1 - \beta_1}} \sum_{j=1}^{N} \mathbb{E}\left[\|U_j\|_2^2\right] \sum_{n=j}^{N} \left(\frac{\beta_1}{\beta_2}\right)^{n-j} \sqrt{1 + n - j}$$

$$\leq \frac{6\alpha_N R}{\sqrt{1 - \beta_1}} \sum_{j=1}^{N} \mathbb{E}\left[\|U_j\|_2^2\right] \frac{1}{(1 - \beta_1/\beta_2)^{3/2}}, \tag{A.41}$$

using Lemma A.3. Finally, using Lemma 5.2 or equivalently Lemma A.2 with $\beta_1 = 0$, we get

$$D \leq \frac{6\alpha_N R}{\sqrt{1 - \beta_1}(1 - \beta_1/\beta_2)^{3/2}} \sum_{i \in [d]} \left(\ln\left(1 + \frac{v_{N,i}}{\epsilon}\right) - N\ln(\beta_2)\right). \tag{A.42}$$

This is as far as we can get without having to use the specific form of $\alpha_N$ given by either (A.2) for Adam or (A.3) for Adagrad. We will now split the proof for either algorithm.

**Adam** For Adam, using (A.2), we have $\alpha_n = (1 - \beta_1)\Omega_n\alpha$. Thus, we can simplify the $A$ term from (A.37), also using the usual change of index $j = n - k$, to get

$$A = \frac{1}{2R} \sum_{n=1}^{N} \frac{\alpha_n}{\Omega_n} \sum_{j=1}^{n} \beta_1^{n-j} \mathbb{E}\left[\|G_j\|_2^2\right]$$

$$= \frac{\alpha(1 - \beta_1)}{2R} \sum_{j=1}^{N} \mathbb{E}\left[\|G_j\|_2^2\right] \sum_{n=j}^{N} \beta_1^{n-j}$$

$$= \frac{\alpha}{2R} \sum_{j=1}^{N} (1 - \beta_1^{N-j+1}) \mathbb{E}\left[\|G_j\|_2^2\right]$$

$$= \frac{\alpha}{2R} \sum_{j=1}^{N} (1 - \beta_1^{N-j+1}) \mathbb{E}\left[\|\nabla F(x_{j-1})\|_2^2\right]$$

$$= \frac{\alpha}{2R} \sum_{j=0}^{N-1} (1 - \beta_1^{N-j}) \mathbb{E}\left[\|\nabla F(x_j)\|_2^2\right]. \tag{A.43}$$

If we now introduce $\tau$ as in (A.8), we can first notice that

$$\sum_{j=0}^{N-1} (1 - \beta_1^{N-j}) = N - \beta_1 \frac{1 - \beta_1^N}{1 - \beta_1} \geq N - \frac{\beta_1}{1 - \beta_1}. \tag{A.44}$$

Introducing

$$\tilde{N} = N - \frac{\beta_1}{1 - \beta_1}, \tag{A.45}$$

we then have

$$A \geq \frac{\alpha\tilde{N}}{2R} \mathbb{E}\left[\|\nabla F(x_\tau)\|_2^2\right]. \tag{A.46}$$

Further notice that for any coordinate $i \in [d]$, we have $v_{N,i} \leq \frac{R^2}{1-\beta_2}$, besides $\alpha_N \leq \alpha\frac{1-\beta_1}{\sqrt{1-\beta_2}}$, so that putting together (A.37), (A.46), (A.38), (A.40) and (A.42) we get

$$\mathbb{E}\left[\|\nabla F(x_\tau)\|_2^2\right] \leq 2R\frac{F_0 - F_*}{\alpha\tilde{N}} + \frac{E}{\tilde{N}}\left(\ln\left(1 + \frac{R^2}{\epsilon(1 - \beta_2)}\right) - N\log(\beta_2)\right), \tag{A.47}$$

with

$$E = \frac{\alpha d R L (1 - \beta_1)}{(1 - \beta_1/\beta_2)(1 - \beta_2)} + \frac{2\alpha^2 d L^2 \beta_1}{(1 - \beta_1/\beta_2)(1 - \beta_2)^{3/2}} + \frac{12 d R^2 \sqrt{1 - \beta_1}}{(1 - \beta_1/\beta_2)^{3/2}\sqrt{1 - \beta_2}}. \tag{A.48}$$

This conclude the proof of theorem 4.

**Adagrad**  For Adagrad, we have $\alpha_n = (1 - \beta_1)\alpha$, $\beta_2 = 1$ and $\Omega_n \leq \sqrt{N}$ so that,

$$
\begin{aligned}
A &= \frac{1}{2R} \sum_{n=1}^{N} \frac{\alpha_n}{\Omega_n} \sum_{j=1}^{n} \beta_1^{n-j} \mathbb{E}\left[ \|G_j\|_2^2 \right] \\
&\geq \frac{\alpha(1 - \beta_1)}{2R\sqrt{N}} \sum_{j=1}^{N} \mathbb{E}\left[ \|G_j\|_2^2 \right] \sum_{n=j}^{N} \beta_1^{n-j} \\
&= \frac{\alpha}{2R\sqrt{N}} \sum_{j=1}^{N} (1 - \beta_1^{N-j+1}) \mathbb{E}\left[ \|G_j\|_2^2 \right] \\
&= \frac{\alpha}{2R\sqrt{N}} \sum_{j=1}^{N} (1 - \beta_1^{N-j+1}) \mathbb{E}\left[ \|\nabla F(x_{j-1})\|_2^2 \right] \\
&= \frac{\alpha}{2R\sqrt{N}} \sum_{j=0}^{N-1} (1 - \beta_1^{N-j}) \mathbb{E}\left[ \|\nabla F(x_j)\|_2^2 \right].
\end{aligned}
\tag{A.49}
$$

Reusing (A.44) and (A.45) from the Adam proof, and introducing $\tau$ as in (9), we immediately have

$$A \geq \frac{\alpha \tilde{N}}{2R\sqrt{N}} \mathbb{E}\left[ \|\nabla F(x_\tau)\|_2^2 \right]. \tag{A.50}$$

Further notice that for any coordinate $i \in [d]$, we have $v_N \leq NR^2$, besides $\alpha_N = (1 - \beta_1)\alpha$, so that putting together (A.37), (A.50), (A.38), (A.40) and (A.42) with $\beta_2 = 1$, we get

$$\mathbb{E}\left[ \|\nabla F(x_\tau)\|_2^2 \right] \leq 2R\sqrt{N} \frac{F_0 - F_*}{\alpha \tilde{N}} + \frac{\sqrt{N}}{\tilde{N}} E \ln\left( 1 + \frac{NR^2}{\epsilon} \right), \tag{A.51}$$

with

$$E = \alpha d R L + \frac{2\alpha^2 d L^2 \beta_1}{1 - \beta_1} + \frac{12 d R^2}{1 - \beta_1}. \tag{A.52}$$

This conclude the proof of theorem 3.

### A.7  Proof variant using Hölder inequality

Following (Ward et al., 2019; Zou et al., 2019b), it is possible to get rid of the almost sure bound on the gradient given by (7), and replace it with a bound in expectation, i.e.

$$\forall x \in \mathbb{R}^d, \ \mathbb{E}\left[ \|\nabla f(x)\|_2^2 \right] \leq \tilde{R} - \sqrt{\epsilon}. \tag{A.53}$$

Note that we now need an $\ell_2$ bound in order to properly apply the Hölder inequality hereafter:

We do not provide the full proof for the result, but point the reader to the few places where we have used (7). We first use it in Lemma A.1. We inject R into (A.15), which we can just replace with $\tilde{R}$. Then we use (7) to bound $r$ and derive (A.26). Remember that $r$ is defined in (A.20), and is actually a weighted sum of the squared gradients in expectation. Thus, a bound in expectation is acceptable, and Lemma A.1 is valid replacing the assumption (7) with (A.53).

Looking at the actual proof, we use (6) in a single place: just after (A.35), in order to derive an upper bound for the denominator in the following term:

$$M = \frac{\alpha_n}{2} \left( \sum_{i \in [d]} \sum_{k=0}^{n-1} \beta_1^k \mathbb{E} \left[ \frac{G_{n-k,i}^2}{2\sqrt{\epsilon + \tilde{v}_{n,k+1,i}}} \right] \right). \tag{A.54}$$

Let us introduce $\tilde{V}_{n,k+1} = \sum_{i \in [d]} \tilde{v}_{n,k+1,i}$. We immediately have that

$$M \geq \frac{\alpha_n}{2} \left( \sum_{k=0}^{n-1} \beta_1^k \mathbb{E} \left[ \frac{\|G_{n-k}\|_2^2}{2\sqrt{\epsilon + \tilde{V}_{n,k+1}}} \right] \right) \tag{A.55}$$

Taking $X = \left( \frac{\|G_{n-k}\|_2^2}{\sqrt{\epsilon + \tilde{V}_{n,k+1}}} \right)^{\frac{2}{3}}$, $Y = \left( \sqrt{\epsilon + \tilde{V}_{n,k+1}} \right)^{\frac{2}{3}}$, we can apply Hölder inquality as

$$\mathbb{E} \left[ |X|^{\frac{3}{2}} \right] \geq \left( \frac{\mathbb{E} \left[ |XY| \right]}{\mathbb{E} \left[ |Y|^3 \right]^{\frac{1}{3}}} \right)^{\frac{3}{2}}, \tag{A.56}$$

which gives us

$$\mathbb{E} \left[ \frac{\|G_{n-k}\|_2^2}{\sqrt{\epsilon + \tilde{V}_{n,k+1}}} \right] \geq \frac{\mathbb{E} \left[ \|G_{n-k}\|_2^{\frac{4}{3}} \right]^{\frac{3}{2}}}{\sqrt{\mathbb{E} \left[ \epsilon + \tilde{V}_{n,k+1} \right]}} \geq \frac{\mathbb{E} \left[ \|G_{n-k}\|_2^{\frac{4}{3}} \right]^{\frac{3}{2}}}{\Omega_n \tilde{R}}, \tag{A.57}$$

with $\Omega_n = \sqrt{\sum_{j=0}^{n-1} \beta_2^j}$, and using the fact that $\mathbb{E} \left[ \epsilon + \sum_{i \in [d]} \tilde{v}_{n,k+1,i} \right] \leq R^2 \Omega_n^2$.

Thus we can recover almost exactly (A.36) except we have to replace all terms of the form $\mathbb{E} \left[ \|G_{n-k}\|_2^2 \right]$ with $\mathbb{E} \left[ \|G_{n-k}\|_2^{\frac{4}{3}} \right]^{\frac{3}{2}}$. The rest of the proof follows as before, with all the dependencies in $\alpha, \beta_1, \beta_2$ remaining the same.

## B  Non convex SGD with heavy-ball momentum

We extend the existing proof of convergence for SGD in the non convex setting to use heavy-ball momentum (Ghadimi & Lan, 2013). Compared with previous work on momentum for non convex SGD byYang et al. (2016), we improve the dependency in $\beta_1$ from $O((1-\beta_1)^{-2})$ to $O((1-\beta_1)^{-1})$. A recent work by Liu et al. (2020) achieve a similar dependency of $O(1/(1-\beta_1))$, with weaker assumptions (without the bounded gradients assumptions).

### B.1  Assumptions

We reuse the notations from Section 2.1. Note however that we use here different assumptions than in Section 2.3. We first assume $F$ is bounded below by $F_*$, that is,

$$\forall x \in \mathbb{R}^d, \ F(x) \geq F_*. \tag{B.1}$$

We then assume that the stochastic gradients have bounded variance, and that the gradients of $F$ are uniformly bounded, i.e. there exist $R$ and $\sigma$ so that

$$\forall x \in \mathbb{R}^d, \|\nabla F(x)\|_2^2 \leq R^2 \quad \text{and} \quad \mathbb{E} \left[ \|\nabla f(x)\|_2^2 \right] - \|\nabla F(x)\|_2^2 \leq \sigma^2, \tag{B.2}$$

and finally, the *smoothness of the objective function*, e.g., its gradient is $L$-Liptchitz-continuous with respect to the $\ell_2$-norm:

$$\forall x, y \in \mathbb{R}^d, \|\nabla F(x) - \nabla F(y)\|_2 \leq L \|x - y\|_2. \tag{B.3}$$

## B.2 Result

Let us take a step size $\alpha > 0$ and a heavy-ball parameter $1 > \beta_1 \geq 0$. Given $x_0 \in \mathbb{R}^d$, taking $m_0 = 0$, we define for any iteration $n \in \mathbb{N}^*$ the iterates of SGD with momentum as,

$$\begin{cases} m_n &= \beta_1 m_{n-1} + \nabla f_n(x_{n-1}) \\ x_n &= x_{n-1} - \alpha m_n. \end{cases} \tag{B.4}$$

Note that in (B.4), the scale of the typical size of $m_n$ will increases with $\beta_1$. For any total number of iterations $N \in \mathbb{N}^*$, we define $\tau_N$ a random index with value in $\{0, \ldots, N-1\}$, verifying

$$\forall j \in \mathbb{N}, j < N, \mathbb{P}\left[\tau = j\right] \propto 1 - \beta_1^{N-j}. \tag{B.5}$$

**Theorem B.1** (Convergence of SGD with momemtum). *Given the assumptions from Section B.1, given $\tau$ as defined in* (B.5) *for a total number of iterations $N > \frac{1}{1-\beta_1}$, $x_0 \in \mathbb{R}^d$, $\alpha > 0$, $1 > \beta_1 \geq 0$, and $(x_n)_{n \in \mathbb{N}^*}$ given by* (B.4),

$$\mathbb{E}\left[\|\nabla F(x_\tau)\|_2^2\right] \leq \frac{1-\beta_1}{\alpha \tilde{N}}(F(x_0) - F_*) + \frac{N}{\tilde{N}} \frac{\alpha L(1+\beta_1)(R^2 + \sigma^2)}{2(1-\beta_1)^2}, \tag{B.6}$$

*with $\tilde{N} = N - \frac{\beta_1}{1-\beta_1}$.*

## B.3 Analysis

We can first simplify (B.6), if we assume $N \gg \frac{1}{1-\beta_1}$, which is always the case for practical values of $N$ and $\beta_1$, so that $\tilde{N} \approx N$, and,

$$\mathbb{E}\left[\|\nabla F(x_\tau)\|_2^2\right] \leq \frac{1-\beta_1}{\alpha N}(F(x_0) - F_*) + \frac{\alpha L(1+\beta_1)(R^2 + \sigma^2)}{2(1-\beta_1)^2}. \tag{B.7}$$

It is possible to achieve a rate of convergence of the form $O(1/\sqrt{N})$, by taking for any $C > 0$,

$$\alpha = (1 - \beta_1)\frac{C}{\sqrt{N}}, \tag{B.8}$$

which gives us

$$\mathbb{E}\left[\|\nabla F(x_\tau)\|_2^2\right] \leq \frac{1}{C\sqrt{N}}(F(x_0) - F_*) + \frac{C}{\sqrt{N}} \frac{L(1+\beta_1)(R^2 + \sigma^2)}{2(1-\beta_1)}. \tag{B.9}$$

In comparison, Theorem 3 by Yang et al. (2016) would give us, assuming now that $\alpha = (1-\beta_1)\min\left\{\frac{1}{L}, \frac{C}{\sqrt{N}}\right\}$,

$$\min_{k \in \{0, \ldots N-1\}} \mathbb{E}\left[\|\nabla F(x_k)\|_2^2\right] \leq \frac{2}{N}(F(x_0) - F_*)\max\left\{2L, \frac{\sqrt{N}}{C}\right\}$$
$$+ \frac{C}{\sqrt{N}} \frac{L}{(1-\beta_1)^2}\left(\beta_1^2(R^2 + \sigma^2) + (1-\beta_1)^2\sigma^2\right). \tag{B.10}$$

We observe an overall dependency in $\beta_1$ of the form $O((1-\beta_1)^{-2})$ for Theorem 3 by Yang et al. (2016), which we improve to $O((1-\beta_1)^{-1})$ with our proof.

Liu et al. (2020) achieves a similar dependency in $(1-\beta_1)$ as here, but with weaker assumptions. Indeed, in their Theorem 1, their result contains a term in $O(1/\alpha)$ with $\alpha \leq (1-\beta_1)M$ for some problem dependent constant $M$ that does not depend on $\beta_1$.

Notice that as the typical size of the update $m_n$ will increase with $\beta_1$, by a factor $1/(1-\beta_1)$, it is convenient to scale down $\alpha$ by the same factor, as we did with (B.8) (without loss of generality, as $C$ can take any value). Taking $\alpha$ of this form has the advantage of keeping the first term on the right hand side in (B.6) independent of $\beta_1$, allowing us to focus only on the second term.

### B.4 Proof

For all $n \in \mathbb{N}^*$, we note $G_n = \nabla F(x_{n-1})$ and $g_n = \nabla f(x_{n-1})$. $\mathbb{E}_{n-1}[\cdot]$ is the conditional expectation with respect to $f_1, \ldots, f_{n-1}$. In particular, $x_{n-1}$ and $m_{n-1}$ are deterministic knowing $f_1, \ldots, f_{n-1}$.

**Lemma B.1** (Bound on $m_n$). *Given $\alpha > 0$, $1 > \beta_1 \geq 0$, and $(x_n)$ and $(m_n)$ defined as by B.4, under the assumptions from Section B.1, we have for all $n \in \mathbb{N}^*$,*

$$\mathbb{E}\left[\|m_n\|_2^2\right] \leq \frac{R^2 + \sigma^2}{(1 - \beta_1)^2}. \tag{B.11}$$

*Proof.* Let us take an iteration $n \in \mathbb{N}^*$,

$$\mathbb{E}\left[\|m_n\|_2^2\right] = \mathbb{E}\left[\left\|\sum_{k=0}^{n-1} \beta_1^k g_{n-k}\right\|_2^2\right] \quad \text{using Jensen we get,}$$

$$\leq \left(\sum_{k=0}^{n-1} \beta_1^k\right) \sum_{k=0}^{n-1} \beta_1^k \mathbb{E}\left[\|g_{n-k}\|_2^2\right]$$

$$\leq \frac{1}{1 - \beta_1} \sum_{k=0}^{n-1} \beta_1^k (R^2 + \sigma^2)$$

$$= \frac{R^2 + \sigma^2}{(1 - \beta_1)^2}.$$

□

**Lemma B.2** (sum of a geometric term times index). *Given $0 < a < 1$, $i \in \mathbb{N}$ and $Q \in \mathbb{N}$ with $Q \geq i$,*

$$\sum_{q=i}^{Q} a^q q = \frac{a^i}{1-a}\left(i - a^{Q-i+1}Q + \frac{a - a^{Q+1-i}}{1-a}\right) \leq \frac{a}{(1-a)^2}. \tag{B.12}$$

*Proof.* Let $A_i = \sum_{q=i}^{Q} a^q q$, we have

$$A_i - aA_i = a^i i - a^{Q+1} Q + \sum_{q=i+1}^{Q} a^q (i + 1 - i)$$

$$(1-a)A_i = a^i i - a^{Q+1} Q + \frac{a^{i+1} - a^{Q+1}}{1-a}.$$

Finally, taking $i = 0$ and $Q \to \infty$ gives us the upper bound. □

**Lemma B.3** (Descent lemma). *Given $\alpha > 0$, $1 > \beta_1 \geq 0$, and $(x_n)$ and $(m_n)$ defined as by B.4, under the assumptions from Section B.1, we have for all $n \in \mathbb{N}^*$,*

$$\mathbb{E}\left[\nabla F(x_{n-1})^T m_n\right] \geq \sum_{k=0}^{n-1} \beta_1^k \mathbb{E}\left[\|\nabla F(x_{n-k-1})\|_2^2\right] - \frac{\alpha L \beta_1 (R^2 + \sigma^2)}{(1 - \beta_1)^3} \tag{B.13}$$

*Proof.* For simplicity, we note $G_n = \nabla F(x_{n-1})$ the expected gradient and $g_n = \nabla f_n(x_{n-1})$ the stochastisc gradient at iteration $n$.

$$G_n^T m_n = \sum_{k=0}^{n-1} \beta_1^k G_n^T g_{n-k}$$

$$= \sum_{k=0}^{n-1} \beta_1^k G_{n-k}^T g_{n-k} + \sum_{k=1}^{n-1} \beta_1^k (G_n - G_{n-k})^T g_{n-k}. \tag{B.14}$$

This last step is the main difference with previous proofs with momentum (Yang et al., 2016): we replace the current gradient with an old gradient in order to obtain extra terms of the form $\|G_{n-k}\|_2^2$. The price to pay is the second term on the right hand side but we will see that it is still beneficial to perform this step. Notice that as $F$ is $L$-smooth so that we have, for all $k \in \mathbb{N}^*$

$$\|G_n - G_{n-k}\|_2^2 \leq L^2 \left\| \sum_{l=1}^{k} \alpha m_{n-l} \right\|^2$$

$$\leq \alpha^2 L^2 k \sum_{l=1}^{k} \|m_{n-l}\|_2^2, \tag{B.15}$$

using Jensen inequality. We apply

$$\forall \lambda > 0, \, x, y \in \mathbb{R}, \|xy\|_2 \leq \frac{\lambda}{2} \|x\|_2^2 + \frac{\|y\|_2^2}{2\lambda}, \tag{B.16}$$

with $x = G_n - G_{n-k}$, $y = g_{n-k}$ and $\lambda = \frac{1-\beta_1}{k\alpha L}$ to the second term in (B.14), and use (B.15) to get

$$G_n^T m_n \geq \sum_{k=0}^{n-1} \beta_1^k G_{n-k}^T g_{n-k} - \sum_{k=1}^{n-1} \frac{\beta_1^k}{2} \left( \left( (1-\beta_1)\alpha L \sum_{l=1}^{k} \|m_{n-l}\|_2^2 \right) + \frac{\alpha L k}{1-\beta_1} \|g_{n-k}\|_2^2 \right).$$

Taking the full expectation we have

$$\mathbb{E}\left[G_n^T m_n\right] \geq \sum_{k=0}^{n-1} \beta_1^k \mathbb{E}\left[G_{n-k}^T g_{n-k}\right] - \alpha L \sum_{k=1}^{n-1} \frac{\beta_1^k}{2} \left( \left( (1-\beta_1) \sum_{l=1}^{k} \mathbb{E}\left[\|m_{n-l}\|_2^2\right] \right) + \frac{k}{1-\beta_1} \mathbb{E}\left[\|g_{n-k}\|_2^2\right] \right). \tag{B.17}$$

Now let us take $k \in \{0, \ldots, n-1\}$, first notice that

$$\mathbb{E}\left[G_{n-k}^T g_{n-k}\right] = \mathbb{E}\left[\mathbb{E}_{n-k-1}\left[\nabla F(x_{n-k-1})^T \nabla f_{n-k}(x_{n-k-1})\right]\right]$$

$$= \mathbb{E}\left[\nabla F(x_{n-k-1})^T \nabla F(x_{n-k-1})\right]$$

$$= \mathbb{E}\left[\|G_{n-k}\|_2^2\right].$$

Furthermore, we have $\mathbb{E}\left[\|g_{n-k}\|_2^2\right] \leq R^2 + \sigma^2$ from (B.2), while $\mathbb{E}\left[\|m_{n-k}\|_2^2\right] \leq \frac{R^2+\sigma^2}{(1-\beta_1)^2}$ using (B.11) from Lemma B.1. Injecting those three results in (B.17), we have

$$\mathbb{E}\left[G_n^T m_n\right] \geq \sum_{k=0}^{n-1} \beta_1^k \mathbb{E}\left[\|G_{n-k}\|_2^2\right] - \alpha L(R^2+\sigma^2) \sum_{k=1}^{n-1} \frac{\beta_1^k}{2} \left( \left( \frac{1}{1-\beta_1} \sum_{l=1}^{k} 1 \right) + \frac{k}{1-\beta_1} \right) \tag{B.18}$$

$$= \sum_{k=0}^{n-1} \beta_1^k \mathbb{E}\left[\|G_{n-k}\|_2^2\right] - \frac{\alpha L}{1-\beta_1}(R^2+\sigma^2) \sum_{k=1}^{n-1} \beta_1^k k. \tag{B.19}$$

Now, using (B.12) from Lemma B.2, we obtain

$$\mathbb{E}\left[G_n^T m_n\right] \geq \sum_{k=0}^{n-1} \beta_1^k \mathbb{E}\left[\|G_{n-k}\|_2^2\right] - \frac{\alpha L \beta_1(R^2+\sigma^2)}{(1-\beta_1)^3}, \tag{B.20}$$

which concludes the proof.

$\square$

**Proof of Theorem B.1**

*Proof.* Let us take a specific iteration $n \in \mathbb{N}^*$. Using the smoothness of $F$ given by (B.3), we have,

$$F(x_n) \leq F(x_{n-1}) - \alpha G_n^T m_n + \frac{\alpha^2 L}{2} \|m_n\|_2^2. \tag{B.21}$$

Taking the expectation, and using Lemma B.3 and Lemma B.1, we get

$$\mathbb{E}[F(x_n)] \leq \mathbb{E}[F(x_{n-1})] - \alpha \left( \sum_{k=0}^{n-1} \beta_1^k \mathbb{E}\left[\|G_{n-k}\|_2^2\right] \right) + \frac{\alpha^2 L \beta_1 (R^2 + \sigma^2)}{(1-\beta_1)^3} + \frac{\alpha^2 L (R^2 + \sigma^2)}{2(1-\beta_1)^2}$$

$$\leq \mathbb{E}[F(x_{n-1})] - \alpha \left( \sum_{k=0}^{n-1} \beta_1^k \mathbb{E}\left[\|G_{n-k}\|_2^2\right] \right) + \frac{\alpha^2 L (1+\beta_1)(R^2 + \sigma^2)}{2(1-\beta_1)^3} \tag{B.22}$$

rearranging, and summing over $n \in \{1, \dots, N\}$, we get

$$\underbrace{\alpha \sum_{n=1}^{N} \sum_{k=0}^{n-1} \beta_1^k \mathbb{E}\left[\|G_{n-k}\|_2^2\right]}_{A} \leq F(x_0) - \mathbb{E}[F(x_N)] + N \frac{\alpha^2 L (1+\beta_1)(R^2 + \sigma^2)}{2(1-\beta_1)^3} \tag{B.23}$$

Let us focus on the $A$ term on the left-hand side first. Introducing the change of index $i = n - k$, we get

$$A = \alpha \sum_{n=1}^{N} \sum_{i=1}^{n} \beta_1^{n-i} \mathbb{E}\left[\|G_i\|_2^2\right]$$

$$= \alpha \sum_{i=1}^{N} \mathbb{E}\left[\|G_i\|_2^2\right] \sum_{n=i}^{N} \beta_1^{n-i}$$

$$= \frac{\alpha}{1-\beta_1} \sum_{i=1}^{N} \mathbb{E}\left[\|\nabla F(x_{i-1})\|_2^2\right] (1 - \beta^{N-i+1})$$

$$= \frac{\alpha}{1-\beta_1} \sum_{i=0}^{N-1} \mathbb{E}\left[\|\nabla F(x_i)\|_2^2\right] (1 - \beta^{N-i}). \tag{B.24}$$

We recognize the unnormalized probability given by the random iterate $\tau$ as defined by (B.5). The normalization constant is

$$\sum_{i=0}^{N-1} 1 - \beta_1^{N-i} = N - \beta_1 \frac{1 - \beta_1^N}{1 - \beta} \geq N - \frac{\beta_1}{1 - \beta_1} = \tilde{N},$$

which we can inject into (B.24) to obtain

$$A \geq \frac{\alpha \tilde{N}}{1 - \beta_1} \mathbb{E}\left[\|\nabla F(x_\tau)\|_2^2\right]. \tag{B.25}$$

Injecting (B.25) into (B.23), and using the fact that $F$ is bounded below by $F_*$ (B.1), we have

$$\mathbb{E}\left[\|\nabla F(x_\tau)\|_2^2\right] \leq \frac{1 - \beta_1}{\alpha \tilde{N}} (F(x_0) - F_*) + \frac{N}{\tilde{N}} \frac{\alpha L (1+\beta_1)(R^2 + \sigma^2)}{2(1-\beta_1)^2} \tag{B.26}$$

$$\tag{B.27}$$

which concludes the proof of Theorem B.1.

$$\square$$

