# OpenReview forum: "A Simple Convergence Proof of Adam and Adagrad"
_TMLR — Accepted by TMLR_

### Review · Reviewer_Z1mR · 2022-08-06

**Summary Of Contributions:**

The work focuses on non-convex smooth stochastic minimization problem with bounded stochastic gradients (almost surely). In this setup, the authors propose new convergence analysis of Adagrad (with/without momentum) and Adam and derived new convergence rates $O(d\ln(N) / \sqrt{N})$ for both methods. Moreover, the derived rates have better dependence on the heavy-ball momentum parameter $\beta_1$ than the previous best-known rates. As a separate contribution of this work one can emphasize the simplicity fo the proposed proofs.

**Broader Impact Concerns:**

I do not have any broader impact concerns.

**Requested Changes:**

### Requests (I think, this changes are necessary)

1. The discussion about the necessity of bounded stochastic gradients assumption is needed. In particular, it will be interesting to see what part of the proof significantly relies on this assumption. Maybe, some simple motivating examples (either theoretical or empricial) could illustrate this phenomenon better. Next, Zou et al. (2019a,b) use bounded second moment assumption in their results. However, they obtain a bound on $\mathbb{E}[\|\nabla F(x_\tau)\|^{4/3}]^{2/3}$, which is weaker than the bound on $\mathbb{E}[\|\nabla F(x_\tau)\|^{2}]$ derive in this work. The authors also point out that it is possible to derive similar results under bounded second moment assumption. Could the authors add the proof for at least one **momentum** method under this assumption? In particular, it would be interesting to confirm whether the dependence on $\beta_1$ will remain the same or it will become as in [Zou et al., 2019a,b].

2. As I mentioned in the weaknesses section, the comparison of rates for momentum SGD with the rates from [Defazio, 2020] and [Liu et al., 2020] is missing. I believe this should be added. It would be great to add the discussion of the differences between the analyses. [Next part is not a strict request, but rather a recommendation] In particular, I am curious: is it possible to modify the analysis of Momentum SGD to get rid of the bounded gradient assumption?

3. In the numerical experiments, I did not get why the method does not converge for $\beta_1 \neq 0$ in Figure 1a: the norm of the gradient is always large. In figure 1b, it seems that the method did not reach good squared norm of the gradient.

### Suggestions (I think, this suggestions may help to strengthen the paper)

1. **On $O(\cdot)$ notation.** Although the authors explicitly mention and discuss the dependence on $d$ (dimension of the problem) in the derived bounds, they do not show this dependence in the rates written using $O(\cdot)$-notation. I think, it is worth to explicitly show this dependence in such cases. Moreover, I think that the authors could also add that $R$ is an upper bound for $\ell_\infty$-norm. When gradients are dense and many components are of the same size, $\ell_\infty$-norm is $\sqrt{d}$-times smaller than $\ell_2$-norm, which is typically used in the estimates. In this case, the rates are proportional to $\sqrt{d}$, in some sense.

2. As I mentioned before, assumption (6) is quite strong especially when the optimization problem is unconstrained. Therefore, the natural question arises: are the proofs generalizable to the constrained case (when one can project to the set where the problem is defined)? If yes, it would be great to generalize them. If no, it would be great to discuss this limitation and point out what part of the proof is unclear ofr the possible generalization.

3. I am also curious whether it is possible to remove the logarithmic factors from the convergence guarantees for Adagrad and Adam without spoiling the rate. Could the authors add the discussion in this?

4. Could the authors test Adam with the parameters used in theory vs Adam with practical default parameters recommended by Kingma & Ba (2015)? It is interesting to see, how Adam performs in practice with this parameters.

5. Is it possible to extend the analysis to cover scalar version of Adagrad and Adam?

### Minor comments

1. If I am not mistaken, in inequality (13), the last term should be multiplied by some numerical constant (e.g., 10 is enough), since $- \ln(1-N^{-b}) \geq N^{-b}$.

2. In inequality (14), one should have $dR^2$ instead of $dR$, right?

3. In inequality (19), there should be a plus sign after the first expectation in the RHS.

4. In the proof of Lemma 5.2, monotonicity of logarithm is used, not concavity, if I understand correctly.

5. In inequality (30), there is extra dot after $u_n$ in the second term in RHS.

6. Figure 1: how do the authors evaluate the expectation $\mathbb{E}[\|F(x_\tau)\|^2]$? Or is it the squared norm for the last iterate?

7. In inequality (A.11), constant $E$ is introduced, which is already defined as $C$ in the main part of the paper.

8. Typos in Appendix A.4: in the last sentence of the first paragraph: "int (A.54)" $\to$ "in (A.54)". In the second paragraph: "we recommand" $\to$ "we recommend"

9. In the line above (A.35), one should have $b_l$ instead of $b_j$ in the end of the derivation.

10. There is an extra space before the line after (B.4).

11. There is an unnecessary repetition of the text after (B.5). This was explained previously in the paper.

12. Theorem B.1: "Assuming the assumptions" -- this phrase does not sound natural.

13. In the line above (B.18), the bound for the second moment of momentum should contain $\beta_1$ instead of $\beta_2$.

**Strengths And Weaknesses:**

### Strengths

1. **Improvements in the dependence on $\beta_1$.** Compared to the previously best-known rates for Adagrad (Zou et al., 2019b) and Adam (Zou et al., 2019a), the rates from this paper have better dependence on momentum parameter $\beta_1$. That is, the derived upper bounds are proportional to $O((1 - \beta_1)^{-1})$, while the previously best-known rates have $O((1-\beta_1)^{-3})$ (Adagrad) and $O((1-\beta_1)^{-5})$ dependence on $\beta_1$. Since $\beta_1$ is typically very small in practice, the improvement is noticeable.

2. **Simple proofs.** I have checked all proofs: they are correct and quite easy to understand. Each step of the proof is clearly explained and motivation why is it needed is also evident. The proofs consist of 3 stages: derivation of the lower bound for the inner product between the direction used by a method and the gradient direction (needed after application of smoothness in the proof), estimation of the sums appeared in the first lemma, and application of smoothness.

3. **Observation "Optimal finite horizon Adam is Adagrad"**. The observation provided in Section 4.3 is very insightful. First, one can easily notice that if $\beta_2$ is independent of $N$, then Theorem 2 and 4 establish the convergence of Adam only to some neighborhood of the solution, *which cannot be reduced via stepsize*. This fact is expected since Adam is known to be non-convergent in general, so, it is good that this phenomenon is reflected in the analysis. However, if $\beta_2$ goes to $1$ when $N$ increases, then the issue vanishes. We know that $\beta_2 = 1$ corresponds to Adagrad, so it is natural to expect that the bound for Adam would transform in the bound of Adagrad in this case once $\beta_2$ is close enough to $1$. And this is true for the derived bounds: bound for Adam (almost) coincides with the bound for Adagrad when $\beta_2 = 1 - 1/N$, where $N$ is the number of iterations. I think this is a very good property of the proposed analysis, since it reflects some basic intuition about the relation between Adagrad and Adam.

4. **Clarity and structure.** The paper has a good structure and is realtively clean: I haven't found many typos or mistakes in English. This fact make the paper easily-readable.

### Weaknesses

1. **Bounded stochastic gradients.** The assumption from (6), i.e., uniformly bounded stochastic gradients, is quite strong, especially when the optimization problem is unconstrained.

2. **Some related works are not discussed.** In particular, Defazio (2020) and Liu et al. (2020) also derive O((1-\beta_1)^{-1}) dependence on $\beta_1$ for momentum SGD, but they did it without assuming that $\|F(x)\| \leq R$, i.e., without the bounded gradients assumption. I think, it is important to add the comparison with these works. It also would be great to discuss the differences between the analysis techniques.

---

References:

Defazio, A. (2020). Momentum via primal averaging: Theoretical insights and learning rate schedules for non-convex optimization. arXiv preprint arXiv:2010.00406.

Liu, Y., Gao, Y., & Yin, W. (2020). An improved analysis of stochastic gradient descent with momentum. Advances in Neural Information Processing Systems, 33, 18261-18271.

---

> ### Author Response · Authors · 2022-09-05
> **Reply to reviewer Z1mR**
>
> We thank you for your reviews and very careful proof reading of our manuscript.
>
> **Regarding novel SGD work:** we thank you for pointing those out. We comment mostly on the work by Liu et al. (2020) as Defazio (2020) doesn’t provide a clear bound. The result from Liu et al. has a similar dependency in $\beta_1$, as the step size $\alpha\sim (1 - \beta_1$ and their Theorem 1 has a term in $O(1 / \alpha)$. Their work requires less strict assumptions. At the time this paper was first released (2 years ago) it didn’t seem necessary to go that far as the results at the time were much worse. We removed our claim from the abstract and updated related work to mention that more recent work could get the same result with weaker assumptions.
>
> **Regarding the Holder based bound:** we added Section A.7 where we provide the main changes to apply to the proof with momentum to derive the new bound. This should be sufficient to prove that the dependency in $\beta_1$ is not changed.
>
> **Regarding numerical experiments with $\beta_1$:** for beta_1 we were initializing far away from the optimum, not near the optimal like the other curves, we have corrected that now.
>
> **O(.) notation:** we changed the notation in the abstract to include the dependency in d. In general we do not change the notation as it is standard practice and most of the time irrelevant to the discussion (e.g. in Section 4.3). Besides, we have already taken special care that the bounds are clearly expressed in the constants of the problem. We agree with you regarding the case where the gradients are well balanced on all dimensions. We added that point to the analysis in Section d.
>
> **Bounded gradient:** we have a discussion in 4.2 on what other papers do, in particular using Holder inequality, or more recently with the affine growth condition. We are not aware of any projected gradient work for non convex optimization. If in the convex case it is possible to set as a condition that the minimum is inside the projection space, it is not really possible to do so in the non convex case, and it is entirely possible that the edge of the projection space would have a gradient leading outward, thus keeping the iterate blocked at a point of non zero gradient.
>
> **Logarithmic factor:** we are not aware of any way of getting rid of those factors and we haven’t seen anything like that in the litterature, except for the work that are based on $\epsilon$ being large, and which boils down to an SGD-like proof, but are of course impractical for the typical values of $\epsilon$ used in practice.
>
> **Adam used here vs Adam in Kingma: ** if you mean the difference due to the corrective term change in Section 2.2, we now provide this experiment on Figure 2. If you mean the comparison between Adam with optimal rate parameter vs. Adam in practice, then there is no need to run experiments, as Adam with the optimal rate parameters is shown to be equivalent to Adagrad, and we know that Adagrad is suboptimal for deep learning training. We provide some insight in Section 4.3: Adam (with the default parameter) is like constant step size SGD, i.e. it doesn’t converge but in practice gives the best results, most likely because it reduces the term in $F(x_0) - F_*$ much faster. Here we show that for some hyper parameters, Adam can converge too, but we do not recommend in any way using those parameters for practical training
>
> **Covering both scalar and diagonal:** The diagonal version is the one used in practice. Previous work, in particular Ward et al. 2019 covered scalar Adagrad, thus there is no real point in covering it again, especially since it is mostly used as a first milestone for adaptive proofs, which are a bit simpler in this case.
>
> **Minor comments:** We have fixed the error in eq (13) to use the proper bound on the log, and then we take its limit. We fixed (14) and (19). We fixed the argument in Lemma 5.2. For figure 1, on the toy task we compute explicitely the average of the past gradients, for CIFAR-10, we use as a proxy the average of the squared norm of the expected gradients computed every epochs, not every iteration. We renamed C -> E in all the paper for consistency. We fixed the other typos.
>
> **References:**
> Adagrad stepsizes: Sharp convergence over nonconvex land-scapes, Ward et al. 2019.

---

> > ### Comment · Reviewer_Z1mR · 2022-09-10
> > **Reply of reviewer Z1mR to authors**
> >
> > I thank the authors for their detailed response and the changes they applied to the paper.
> >
> > **Projected gradient in the non-convex case.** It is possible to analyze projected SGD in the non-convex case, but one needs to consider a different convergence criterion: instead of the (expected squared) norm of the gradient one needs to bound the norm of the gradient mapping, see [1, 2] for the analysis of Prox-SGD and [3] for the analysis of Prox-Adam.
> >
> > In general, I believe my requests were properly addressed by the authors.
> >
> > ---
> > [1] Ghadimi, S., Lan, G., & Zhang, H. (2016). Mini-batch stochastic approximation methods for nonconvex stochastic composite optimization. Mathematical Programming, 155(1), 267-305.
> >
> > [2] Davis, D., & Drusvyatskiy, D. (2019). Stochastic model-based minimization of weakly convex functions. SIAM Journal on Optimization, 29(1), 207-239.
> >
> > [3] Yun, J., Lozano, A. C., & Yang, E. (2020). A general family of stochastic proximal gradient methods for deep learning. arXiv preprint arXiv:2007.07484.

---

### Review · Reviewer_yRem · 2022-08-07

**Summary Of Contributions:**

This paper provides analysis of a family of first-order optimization algorithms. The family is parametrized by $\beta_1$ and $\beta_2$, the two exponentially-weighted moving average parameters in Adam, and so includes both Adam and AdaGrad. I believe it also includes RMSProp. Theorems are provided for smooth stochastic losses that bound the sum of gradient magnitude squared in expectation. By appropriate settings of the parameters, it is possible to achieve the optimal $O(1/T^{1/4})$ convergence rate for gradient norm. Some experiments are provided to verify that the gradient indeed becomes small on certain problems.

The results tighten some dependencies on $1-\beta_1$ from previous works, and appear to be a bit shorter and general.


**Requested Changes:**

A discussion (or even better, a resolution) of the above issues would help readers see the nuances in the bounds presented in this paper.

On page 6 equation 19, I believe you mean to add rather than multiply the expectations.


**Strengths And Weaknesses:**


Strengths:
The proofs are not too complicated, and streamline previous analyses to achieve the better $1-\beta_1$ bounds. The proofs seem correct and well presented. The improved dependencies are not so surprising given the similar results in convex settings, or even with slightly different algorithms in the non-convex setting (e.g.  https://arxiv.org/pdf/2007.14294.pdf), but I think the authors are correct in noting that such dependencies have not appeared previously for these specific algorithms, which may be of interest.

Weaknesses:
These weaknesses are more along the lines of things where the paper seems to fall short of its potential:

The Adam algorithm analyzed is not actually Adam as it removes some of the “bias correction” terms. Now, the authors state (and I intuitively agree) that these bias correction terms likely are irrelevant because they decay exponentially fast. However, since they do decay so fast, I feel that there is something missing to make the story complete: if we believe that the provided updates are nearly the same as the original updates, can we just bound the difference in these updates to obtain a bound for the original updates? If not, then perhaps it is a bit premature to claim that the bias correction does not do anything.

I think a truly unified analysis would smoothly interpolate between Adagrad and Adam. As currently written, it seems there is no such smooth interpolation because a constant $\alpha$ is forced at every round. If one instead allowed for $\alpha = 1/\sqrt(n)$, for example, then with $\beta_2=1$, $\beta_1=0$, the “Adam” algorithm would become AdaGrad (because $\lim_{x\to 1} \sqrt{(1-x^n)/(1-x)} = \sqrt{n}$, so $\alpha_n=1$).
However, even with a constant $\alpha$, it seems like the analysis is a bit loose on this edge case: for $\alpha = 1/\sqrt{N}$, $\beta_2=1$, $\beta_1=0$, we obtain $\alpha_n = \sqrt{n}/\sqrt{N}$. Now, intuitively we should expect $v_n = \Theta(n)$ so that the algorithm is SGD with an effective learning rate set to the standard value of $O(1/\sqrt{N})$. Thus, we should expect convergence at the optimal rates. However, the current bound appears to blow up. The remarks at the beginning of section 4.3 seem to try to address this by setting $\beta_2=1-1/N$ and showing that this is an acceptable value. However, I find this to be actually just emphasizing the weirdness of the result: a difference of only $O(1/N)$ in $\beta_2$ actually causes an enormous difference in the bound!

---

> ### Author Response · Authors · 2022-09-05
> **Reply to yRem**
>
> We thank you for your review and comments.
>
> **Regarding the bias corrections terms,** we would like to emphasize that we drop only one of the two, the correction to the momentum, not the one for the average of the squared gradients (i.e. the denominator). The former typically decays in ~10 iterations, while the latter decays in ~1000 iterations. Besides, removing the momentum corrective term is equivalent to having a learning rate warmup, which is intuitively harmless for optimization. On the other hand, removing the second term would lead to multiplying the effective learning rate during the first ~1000 epochs by 1 / sqrt(1 - beta_2), which would be around 30 for typical values of beta_2, which intuitively seem worse. This can be large enough to destabilize early training. We rephrased to make clear that we kept the most important bias correction, in order to simplify the proof. We also added an experiment (see Figure 2) where we compare on CIFAR-10 the dynamics with the original Adam vs. when dropping either one of the corrective terms. We observe no difference when dropping the corrective term on the momentum, while the dynamics is strongly impacted when dropping the corrective term on the denominator, thus confirming our intuition.
>
> **Regarding your second point.** First we show in Section 4.3 that taking $\alpha = 1/\sqrt{N}$ and $\beta_2 = 1 - 1/N$ leads to the same optimal rate of convergence as Adam, thus motivating our observation that with those hyper parameters Adam is equivalent to Adagrad. This is not a strict equivalence, as the type of averaging is a bit different, but lead to the same rate nonetheless.
>
> Regarding taking $\beta_2 = 1$, and $\alpha = 1/ \sqrt{N}$ in the Adam result, the blow up is an artifact of the proof: we neglect terms of the form $1 - \beta_2^n$ as those have little impact for any value of $\beta_2 < 1$. In particular, p.10: “Given that $\beta_2 < 1$, we have $\alpha_n \leq \frac{\alpha}{\sqrt{1 - \beta_2}}$”, while taking the limit as you do would require us to keep a $\sqrt{1 - \beta_2^n}$ term.
>
> This is however not really worth the effort as it is more straightforward to handle the cast $\beta_2 = 1$ separately, which we do in the present paper. Thus we do not believe this is a limitation of the paper, especially since as mentioned before, taking $\beta_2 = 1 - 1/N$ and $\alpha = 1/\sqrt{N}$ gives the optimal rate desired.

---

### Review · Reviewer_BBkw · 2022-08-12

**Summary Of Contributions:**

This submission provides a new convergence analysis for the Adagrad and Adam algorithms.
For Adagrad and Adam without momentum, the proof is simple---less than two pages in length--and intuitive.
The main challenge is to control the amount any step can vary from a descent direction, which the authors do by building on theoretical tools in the literature.
A more complex and lengthy analysis is provided for Adagrad and Adam with momentum;
the resulting convergence rates improve the dependence on the momentum term compared to existing results, but the rate is still slower with momentum than without.
The submission concludes with experiments comparing the predicted dependencies of the convergence rate on step-size and momentum parameters to empirical effects.


**Broader Impact Concerns:**

This is a theoretical paper with no specific ethical implications. The Adam and Adagrad algorithms are already widely used, so studying their convergence does not present any issues.

**Requested Changes:**

I think the following changes would be appropriate:

1. Please clarify the connection between the variant of Adam analyzed and the original method proposed by Kingma and Ba.
2. Try to resolve the forward references in Sections 4.2 and A.3.
3. Clarify the use of 'optimal' in Section 4.3.

**Strengths And Weaknesses:**

Overall, I think this is a strong submission which is well suited to the TMLR format.
The authors provide new technical results on Adam and Adagrad which will be of use to other researchers in optimization.
I recommend that the submission be accepted.

### Correctness

I checked the proofs and the submission is theoretically sound as far as I am able to determine.
All theorems are supported by rigorous proofs which, aside from one minor issue (see specific comments below), appear to be correct.
The experiments are well-executed and sufficient details are provided for replication.

#### Detailed Comments

**Equation 13**: I don't see how obtain the last $N^{-b}$ term in this equation.
    Substituting in the value for $\beta_2$ gives $-\log(1 - N^{-b})$, which, using bounds on the logarithm,
    satisfies
    $$N^{-b} \leq -\log(1 - N^{-b}) \leq \frac{N^{-b}}{1 - N^{-b}},$$
    which gives the slightly worse bound of $1/(N^b - 1)$.
    Can the authors clarify this please?

**Bounds for Convex Functions**: One of the interesting aspects of these results is that they do not require a bounded diameter assumption.
    The standard analysis for Adagrad assumes $\sup_{x \in \mathcal{X}} \|x - x^*\|_2 \leq D$ and then rescales the step-size by $D$ to guarantee convergence.
    Do you think the approach here could be utilized to avoid such an assumption in a convergence proof for convex $f$?


**Equations 4**: This choice of step-size is equivalent to choosing $m_{n} = (1 - \beta_1) \beta_1 m_{n-1} + (1 - \beta_1) \nabla_{i} f(x_n - 1)$, which is not the exponential moving average used in Adam.
    You would need to scale as $\beta_1 / (1 - \beta_1)$ in Equation 1 to obtain Adam by the rescaling in Equation 4.
    This different is most pronounced for $\beta_1 \approx 1$, for which scaling of $m_{n-1}$ is approximately zero in the update compared to Adam, which would use $\beta_1$.
    The effect is least noticeable for $\beta_1 \approx 1/2$, for which it is merely divided by two.
    I think that (1) this should be clarified in the text, which currently states "our algorithm differs from Adam only for the first 50 iterations" and (2) can the authors please clarify the impact of this difference on the algorithm/analysis?

**Effect of Bias Correction**: Again with respect to the statement "our algorithm differs from Adam only for the first 50 iterations.", I would like to note that this may change the entire optimization path.
    So, it is somewhat misleading to remark on the similarity to Adam after initial iterations are complete.

**Equation 8**: Can the authors remark on as to why the sampling distribution for iterates should so heavily discount the most recent iterations?
    This is quite unintuitive to me.

**Section 4.3**: In what sense is the word 'Optimal' being used here?
    It's not clear from the discussion that the choices of $\alpha = N^{-a}$ and $\beta_2 = 1 - N^{-b}$ minimize the upper bound from Equation 10.
    The choices for $a,b$ may be optimal given the form of $\alpha$ and $\beta_2$, but this does not imply optimality of the overall choice.

### Interest

The convergence of Adam is a topic of enduring interest to the optimization for machine learning community.
Although the convergence theorems in this work are not highly novel, the short and elegant proofs for the momentum-free case are a strong contribution and I am confident this submission will be of interest to many researchers.

### Strengths and Weaknesses:

**Strengths**:
- The proofs for Adam and Adagrad without momentum are simple, short, and insightful. I was interested to see that the key element is controlling the deviation from a descent direction.
- The momentum bounds recover the bounds without momentum up to a constant factor when $\beta_1 \rightarrow 0$.
- The text is generally well written with comprehensive related work.

**Weaknesses**:
- It's not clear that the version of Adam analyzed directly recovers the original formulation given by Kingma and Ba.
- The organization of the paper sometimes uses forward references to technical lemmas/equations.


### Minor Comments:

- Page 1: "The best know bounds for Adagrad" --  typo.
- Eq 6: Missing a comma here. A 'quad' between the quantification and the expression would make the expression clearly as well.
- Section 4.2: It is awkward to discuss Lemma 5.2 before the result is introduced in the text. Such forward references should always be avoided if possible.
- Equation 19: Missing a '+' between the two expectations.
- A.1: Using $G_n = \nabla F(x_{n-1})$ and $g_n = \nabla f_n(x_{n-1})$ goes against standard notation in probability, where capital letters denote random variables and lower-case letters denote deterministic quantities.
- Theorem A.1 and A.2: The 'thmtools' package provides a 'restatable' environment which can be used to avoid creating new theorem statements.
- Theorem A.1 The constant was denoted $C$ in the main paper, but here it has changed to $E$.
- A.4: Again, the forward reference to Eq. A.19 should be avoided if possible.

---

> ### Author Response · Authors · 2022-09-05
> **Reply to Reviewer BBkw**
>
> We thank the reviewer for the careful reading of our manuscript.
>
> **Equation 13.** We have fixed the error in equation 13, keeping the exact term in (13) and using the first order development for the approximate upper bound just after. We also corrected in (14), with $+ N / (N -1)$ instead of $+ 1$.
>
> **Bounds for Convex functions:** We believe the proof could be extended to obtain a bound on the primal sub optimality $F(x_n) - F_*$ for the convex case instead of a bound on the squared norm of the gradients. One important aspect is that the proof still requires a uniform bound on the gradients, limiting its scope. It would still apply for problems like smoothed L1 regression or logistic regression without L2 penalty, for which the original Adagrad proof would require the extra bounded diameter assumption. We added this point to the discussion in Section 2.3.
>
> **Regarding eq. 4:** The recurrent formula you are giving is not equivalent to the one presented in eq. (1) and (4) (the formula you provide would lead to a $(1 - \beta_1)^n$ term which is not the same as the single $(1 - \beta_1)$ factor in the learning rate). We have clarified the derivation in Section 2.2, and we hope this lift any doubt on the formulation we are using.
>
> As stated in the paper, there is no difference after a few times $1 / (1 - \beta_1)$ iterations. We agree that in a non convex problem, the first few iterations might lead to completely different solutions. However, we reason that the absence of the corrective term for the bias is equivalent to a warmup of the learning rate, which is common practice for training DNN.
>
> We clarified Section 2.2, by carefully introducing the original Adam step size with corrections and studying each term in turn. We further validate our decision by comparing the training trajectories on CIFAR-10 by removing either of the corrective terms, and showing that dropping the corrective term on the momentum has no observable effect for the value typically used in practice, while the dropping the corrective term on $v_n$ strongly change both the training loss and the norm of the gradient (See Figure 2).
>
>
> **Eq 8: why we shouldn’t sample the last iterates ?**
> This can easily be understood when taking the limit $\beta_1\rightarrow 1$. Given an iteration $n$, the last gradient $\nabla F(x_n)$ has almost no influence on next iterate $x_{n+1}$, as the momentum updates very slowly, and therefore it has almost no influence on $F(x_{n+1})$. Non convex convergence proofs are based on the idea that for each iteration, either the gradient is small, or the primal objective F(x) has decreased by some amount. This is not verified *immediately* with large momentum values, as the latest gradient could be very large, yet the primal value would take some numbers of iterations to decrease due to the delay of the effect of this last gradient due to the momentum. For a large enough number of iterations, whether to sample or not those last iterations will in any way have a limited impact. We added this explanation to the paper in Section 4.2.
>
> **Section 4.3, Optimality**
>
> We have clarified in the paper what we mean by optimality. We define optimality as finding the largest number $q$ such that the bound we obtain is $O(\ln(N)/ N^q)$ for some hyper-parameters $\alpha(N)$ and $\beta_2(N)$. Looking at the bound, we immediately have that convergence can only be achieved if $\alpha(N) \rightarrow 0$ and $\beta_2 \rightarrow 1$ for $N \rightarrow \infty$. This motivates us to assume that both expressions have an asymptotic development in $N\rightarrow \infty$. We believe this kind of analysis is standard in optimization, see in particular [Bach and Moulines 2011] and reference therein.
>
> **Typos**
> We have fixed the typos in the manuscript.
>
> Regarding $G_n$ vs. $g_n$ those are chosen to be consistent with $F$ (the overall objective function) and $f$ (the stochastic estimate). These notations are consistent with a number of previous papers in optimization, in particular (Ward et al. 2018). To prevent any confusion we repeated what each one referred to (expected vs. stochastic gradient).
>
> We switched to using the restatable env. We renamed C -> E in the first theorems, for consistency with the proof in the supplementary.
>
> **Regarding the forward references,** we agree it is best to avoid them. However, we also think it is most important to discuss the results before diving into the details. In the main paper we have rephrased to avoid them entirely when discussing the bounded gradient assumption, and when discussing the dependency in $d$, we first discuss without forward references before diving into the details. Similarly, we added a broad overview in Section A.4 in order to give some context without forward references.

---

> > ### Comment · Reviewer_BBkw · 2022-09-07
> > **Thanks for Response**
> >
> > Thanks for the detailed response.
> >
> > I'm afraid I don't follow your remarks about Eq. 4. --- hopefully we can resolve my confusion. The update for exponential moving averages of the mean and variance used in Adam is
> > $$
> > \begin{aligned}
> > m_{n, i} &= \beta_1 m_{n-1, i} + (1 - \beta_1) \nabla_i f(x_{n-1}) \\\\
> > v_{n, i} &= \beta_2 v_{n-1, i} + (1 - \beta_2) (\nabla_i f(x_{n-1}))^2.
> > \end{aligned}
> > $$
> >
> > If we use the updates as you have written it, we obtain
> >
> > $$
> > \begin{aligned}
> > x_{n, i} &= x_{n-1, i} - \alpha_n \frac{m_{n,i}}{\sqrt{\epsilon + v_{n,i}}}\\\\
> >             &= x_{n-1, i} - \alpha (1-\beta_1) \sqrt{\frac{1 - \beta_2^n}{1-\beta_2}}  \frac{m_{n,i}}{\sqrt{\epsilon + v_{n,i}}} \\\\
> >             &= x_{n-1, i} - \alpha (1-\beta_1) \sqrt{\frac{1 - \beta_2^n}{1-\beta_2}}  \frac{\beta_1 m_{n-1, i} + \nabla_i f(x_{n-1})}{\sqrt{\epsilon + v_{n,i}}} \\\\
> >             &= x_{n-1, i} - \alpha \sqrt{\frac{1 - \beta_2^n}{1-\beta_2}}  \frac{(1-\beta_1) \beta_1 m_{n-1, i} + (1 - \beta_1) \nabla_i f(x_{n-1})}{\sqrt{\epsilon + v_{n,i}}},
> > \end{aligned}
> > $$
> >
> > which looks like a modified version of the Adam update using
> >
> > $$
> > \tilde m_{n, i} = \beta_1 (1 - \beta_1) m_{n-1, i} + (1 - \beta_1) \nabla_i f(x_n)
> > $$
> >
> > as the mean estimate. Note that the same issue applies to the variance estimate and that this has nothing to do with the bias correction. You say in the response that there is no difference aside from bias correction, so what am I missing here?

---

> > > ### Comment · Reviewer_BBkw · 2022-09-07
> > > **I got it now!**
> > >
> > > I looked into it a bit more closely and I agree that the formulations are equivalent. Thanks!

---

> > > > ### Author Response · Authors · 2022-09-08
> > > > **sounds good**
> > > >
> > > > Sounds good, when we updated the paper we tried to make the derivation clearer, especially around eq (4) in the new paper. Let us know if we should refine that.

---

> > > > > ### Comment · Reviewer_BBkw · 2022-09-08
> > > > > **Re: Clarifying the Derviration**
> > > > >
> > > > > As I wrote above, my main confusion was the $\beta_1 (1-\beta_1)$ term multiplied against $m_{n-1}$ --- this makes it _look_ like the history is being decremented by an additional $1 - \beta_1$ factor. What I didn't realize at first was that $m_{n-1}$ is not the same quantity as defined in the original Adam paper; it is exactly that quantity divided by $1 - \beta_1$. Thus, the extra factor can be merged with this term to see the equivalence.
> > > > >
> > > > > The way I saw this was writing down the mean estimate (as given in Adam) as a sum like the following:
> > > > >
> > > > > $$
> > > > > \begin{aligned}
> > > > > m_{n}
> > > > > &= \sum_{i = 1}^n (1 - \beta_1) \beta_1^{n - i} \nabla f(x_i) \\\\
> > > > > &= (1 - \beta_1) \sum_{i = 1}^n \beta_1^{n - i} \nabla f(x_i) \\\\
> > > > > &= (1 - \beta_1) \tilde m_{n},
> > > > > \end{aligned}
> > > > > $$
> > > > >
> > > > > where $\tilde m_{n}$ is the mean estimate as given in this work.
> > > > > This derivation is what allowed me to see the equivalence clearly.

---

> > > > > > ### Author Response · Authors · 2022-09-28
> > > > > > **updated the paper**
> > > > > >
> > > > > > We have updated the paper to include the proposed derivation. We separate the treatment of the $(1 - \beta_1)$ term, made to give a weighted average instead of a weighted sum, and the $(1 - \beta-1^n)$ term, made to remove the bias in the weighted average.

---

### Review · Reviewer_8u7Z · 2022-08-22

**Summary Of Contributions:**

The paper proposes a "simple" proof for Adam and Adagrad for smooth objective functions with bounded gradients. The link between Adam and Adagrad is shown via this proof, along with experimental results with a toy problem and CIFAR10 dataset. The paper has the potential to be impactful for researchers who are interested in proofs that are applicable to other related algorithms as well

**Requested Changes:**

-  [Minor] Without much context, factors in the abstract such as $\beta_1$ does not make sense for most readers (have to scroll down to sec. 2.2); Better to keep the details in the main text and make the abstract lighter, or introduce in abstract for completeness as decay rates for moment estimates.
- [Minor] hyphenate non-convex throughout (also non-adaptive)
- [Minor] Please explain this further or repeat part of (Yang et al., 2016) here - "improves the dependency on $1 − \beta_1$ from $a −2$ to $a −1$"
- [Minor] "rate at which the scale of past gradients is forgotten" can be rephrased. Same comment on "it has a slower forgetting of.." on page 6
- "leads to an algorithm close to Adam" - this is important to note, and "for a typical $\beta_1$ of 0.9 ... 50 iterations", seems like a justification to simplify the proof. This is no longer Adam, but perhaps the paper can be scoped down to a proof for Adagrad. Alternatively, provide the more complicated proof for full Adam as supplementary material.
- [Minor] References for "Image generation, music synthesis and Language modeling" seem like fillers. Can you simplify this paragraph? Remove unnecessary references and rephrase sentences like "does Adam really not converge?"
- Would be better to show general proof, and then apply $\beta_1 = 0$; this is just a suggestion, feel free to explain why this ordering is better for you. In fact, are theorem 3 and 4 sufficient? -- this is important to discuss on this topic - "Taking $\beta_1 = 0 $ in those theorems gives us **almost exactly** the bound without heavy-ball momentum from Theorems 1 and 2"
- recheck Lemma 5.1 for correctness
- [Minor] "weights are chosen so that the variances of all the coordinates of the gradient are equals1" -- "equal"? fullstop?
- Bounded gradient assumption needs more discussion. This is a very strong assumption and may not apply to most cases. In fact, this may be where your CIFAR 10 experiment varies from theory. Also, the toy problem does not add much value, can you consider another experiment?




-

**Strengths And Weaknesses:**

## Strengths

+ Recovering previous results such as convergence rate for Adam is good
+ Common notation is useful for the community to take forward
+ Improving tightness around $\beta_1$ is a much needed improvement over older proofs
+ In general proofs are well written



## Weaknesses
- The deviation from the original Adam, and simplified proofs depends on simplification of the algorithm itself.
- Some assumptions are strong, and need to be justified in the text / as part of the proof

---

> ### Author Response · Authors · 2022-09-05
> **reply to reviewer 8u7Z**
>
> We thank Reviewer 8u7Z for their review.
>
> We have removed references to $\alpha$ and $\beta_2$ in the abstract. We kept the reference to $\beta_1$, after having introduced it as the momentum decay rate.
>
> We rephrased “rate at which the scale of past gradients is forgotten” to “controls the decay rate of the per-coordinate exponential moving average of the squared gradients.”
>
> “weights are chosen so that the variances of all the coordinates of the gradient are equals”: we rephrased to “such that all coordinates of the gradient have the same variance.”
>
> **Regarding showing the general proof first:** the use of momentum complexifies the proof, taking it from under two pages to over 8 pages. Thus we defer the more general proof to the supplementary material. We keep the two sets of theorems (with and without momentum) for consistency. The ones without momentum are slightly tighter and easier to read.
>
> **Boundness of the gradients:** we discuss the boundedness of the gradient in Section 4.2. We have further extended this discussion. Recent work such as [Faw et al. 2022] use instead an affine growth assumption of the variance. However, this considerably complexifies the proof technique. We are not claiming here to offer the proof with the weakest assumptions, and present instead a simpler technique.
>
> **Experiments:** the toy experiment is built specifically to show the worst case dependency in $1/\sqrt(1 - \beta_2)$. On most real life problems, this part of the bound rarely dominates, as is shown by the experiment on CIFAR-10 and the common use of values for $\beta_2$ ranging from 0.9 to 0.999.

---

### Decision · Action_Editors · 2022-09-23

**Recommendation:** Accept as is

**Comment:**

The paper as the reviewers note provides a simple and intuitive proof for the convergence of Adam and Adagrad. As the reviewers the analysis will be useful to the community and streamlines previously existing analyses and improves the dependence on 1-beta. The reviewers raised some points and they were answered and appropriate edits were made by the authors. I recommend acceptance.

---

> ### Author Response · Authors · 2022-10-17
> **camera ready provided**
>
> Thank you very much for your time on this paper and accepting it to TMLR. I have updated the camera ready. One thing I wasn't sure was whether the "Appendix" section was allowed inside the main pdf of the paper or should be separated. I think for the reader it is more convenient to have everything together, but if the policy is to have it separated, I can separate it as well.